# Context-Driven Incremental Compression for Multi-Turn Dialogue Generation

Yeongseo Jung [1 2 †]  Jaehyeok Kim [1]  Eunseo Jung [1 2 †]  Jiachuan Wang [3]  Yongqi Zhang [4]  Ka Chun Cheung [2]
Simon See [2]  Lei Chen [1 4]

## Abstract

Modern conversational agents condition on an ever-growing dialogue history at each turn, incurring redundant attention and encoding costs that grow with conversation length. Naive truncation or summarization degrades fidelity, while existing context compressors lack cross-turn memory sharing or revision, causing information loss and compounding errors in long dialogues. We revisit the context compression under conversational dynamics and empirically present its fragility. To improve both efficiency and robustness, we introduce Context-Driven Incremental Compression (C-DIC), which treats a conversation as interleaved contextual threads and stores revisable per-thread compression states in a single, compact dialogue memory. At each turn, a lightweight retrieve → revise → write-back loop shares information across turns and updates stale memories, stabilizing long-horizon behavior. In addition, we adapt truncated backpropagation-through-time (TBPTT) to our multi-turn setting, learning cross-turn dependencies without full-history backpropagation. Extensive experiments on long-form dialogue benchmarks demonstrate superior performance and efficiency of C-DIC; notably, C-DIC shows stable inference latency and perplexity over hundreds of dialogue turns, supporting a scalable path to high-quality dialogue modeling.

†Work during an internship at NVIDIA. [1]Department of Computer Science and Engineering, The Hong Kong University of Science and Technology, Hong Kong [2]NVIDIA AI Technology Center, NVIDIA [3]University of Tsukuba, Japan [4]Thrust of Data Science And Analytics, The Hong Kong University of Science and Technology (Guangzhou), China. Correspondence to: Jiachuan Wang <wangjc@slis.tsukuba.ac.jp>.

*Proceedings of the 43rd International Conference on Machine Learning*, Seoul, South Korea. PMLR 306, 2026. Copyright 2026 by the author(s).

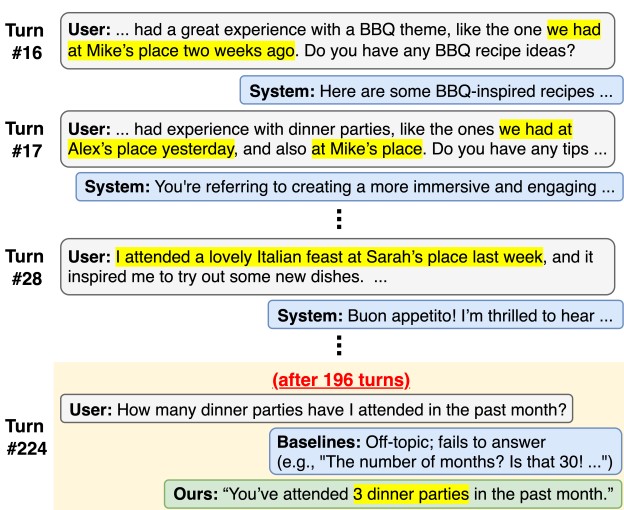

*Figure 1.* **Long-horizon reference tracking.** After 196 intervening turns, the user asks how many dinner parties they attended in the past month. Baselines fail to recover earlier mentions, while C-DIC retrieves thread-relevant memories and answers correctly.

## 1. Introduction

Conversational agents powered by large language models (LLMs), such as CHATGPT (OpenAI, 2022) and Gemini (Gemini Team, 2025), have emerged as ubiquitous interfaces for a wide range of tasks such as brainstorming, code debugging, and data analysis (Yi et al., 2024; Nijkamp et al., 2023). These interactions are characteristically *multi-turn*, where even casual sessions often span dozens of exchanges with topic drifts, cross-turn references, and iterative refinements (Xu et al., 2022). Such interactive adaptability of LLM-based assistants constitutes a pivotal cornerstone of their efficacy and enables capabilities beyond static search or form-based interfaces.

Despite strong single-turn performance, current LLMs struggle to manage the dependencies and drift that arise in multi-turn discourse (Laban et al., 2025). The prevalent naive approach, concatenating the entire conversation history to the prompt at every step, introduces two core challenges. First, it induces significant **computational inefficiency**: repeatedly *re-encoding* and *re-attending* to the full dialogue history at each turn incurs high inference-time costs, as self-attention scales quadratically with input length (Vaswani

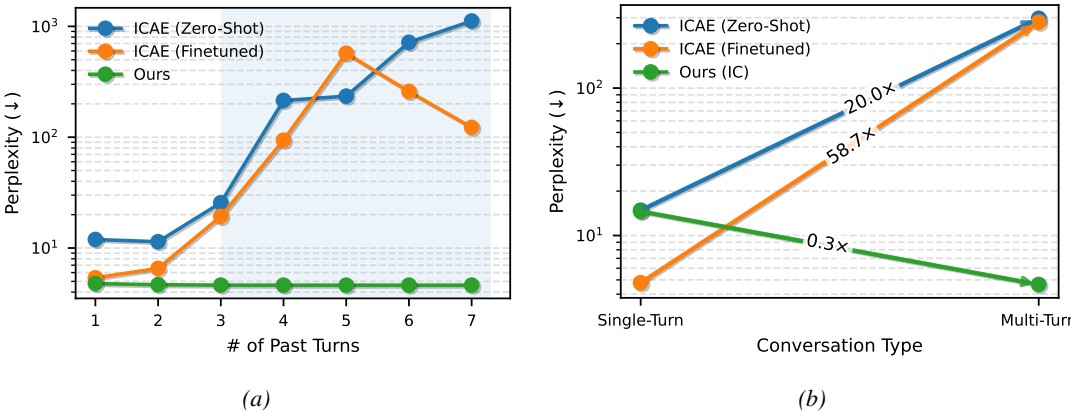

*(a)*              *(b)*

*Figure 2.* **Static compression collapses under multi-turn conversation; C-DIC remains stable.** Perplexity ($\downarrow$) for ICAE (zero-shot), ICAE (MSC-tuned), and our method. **(a)** Static baselines rise sharply after 3-4 consecutive compressions. **(b)** Moving from single-turn (one-shot) to multi-turn evaluation, perplexity for static models explodes by *at least* $\sim 1900\%$ while C-DIC decreases by 70%.

et al., 2023; Tay et al., 2022). Second, it triggers **semantic drift and contextual erosion**: as dialogues evolve, models often *lose the thread*, producing irrelevant responses (Laban et al., 2025). These challenges stem from the model's insufficient focus on dialogue turns that align with the user's evolving intents, especially when such turns lie beyond the model's recency-driven attention scope. For example, as shown in Figure 1, a query may require aggregating scattered mentions across hundreds of intervening turns, where standard baselines fail to reason over the relevant evidence.

Existing methods address efficiency by truncating history to recent turns (Xu et al., 2022; Laban et al., 2025) or by using static summaries (Wang et al., 2025; Packer et al., 2024). Truncation discards long-range dependencies, while static summaries tend to be query-agnostic, lossy, and inflexible for mid-conversation revision (Laban et al., 2025; Ravaut et al., 2024). As a result, these methods frequently degrade coherence and adaptability in dynamic multi-turn settings.

On the other hand, a line of work compresses long static documents into a small set of latent vectors (Chevalier et al., 2023; Ge et al., 2024) for efficiency. However, static, single-shot compressors are brittle under multi-turn rollout: performance degrades substantially under accumulated compression across consecutive dialogue turns, as illustrated in Figure 2. This clearly presents the core limitation of the static compressors lacking mechanisms for memory revision and sharing across consecutive turns.

To address these limitations, we take a principled approach: progressive, topic-aware inference-time compression of the dialogue history, thereby preserving both efficiency and coherence in multi-turn dialogues. In particular, the model should retrieve and reason over context that is semantically aligned with the current topic, regardless of its position in the history. In the absence of such topic sensitivity, even compressed inputs risk overlooking essential context, yielding incoherent or irrelevant responses.

We implement the above principled approach by introducing **Context-Driven Incremental Compression (C-DIC)**, which views a conversation as interleaved contextual threads and maintains a compact dialogue memory of *revisable per-thread compression states*. This setting introduces three technical challenges: topic-aligned retrieval under drift, incremental revision without re-encoding the full history, and efficient inference-time memory management.

C-DIC addresses these challenges with a lightweight *retrieve $\rightarrow$ revise $\rightarrow$ write-back* loop executed at each turn, enabling cross-turn sharing and correction. Unlike static one-shot compression, this loop treats compressed dialogue states as persistent, revisable latent memory rather than as a fixed summary of prior context. **(i) Thread-aware memory retrieval.** At each turn, the model dynamically retrieves the subset of compressed history relevant to the active thread, irrespective of position in the history. **(ii) Incremental compression.** It compresses the current turn with its thread states, allowing future turns to reuse it without re-encoding the full history. **(iii) Gradient-free memory update.** To accommodate evolving and revisited topics, C-DIC performs memory updates online without inference-time gradients. Training mirrors this loop via turn-level, retrieval-aware truncated backpropagation through time, avoiding full-history backpropagation and mitigating error accumulation typical of static, one-shot compressors (Chevalier et al., 2023; Ge et al., 2024). Together, these components form a *topic-sensitive* compression framework that retains salient information while discarding irrelevant content as user intent evolves; using a 7B-scale backbone throughout, C-DIC shows *stable inference latency and perplexity* over **hundreds of turns**.

Our contributions are as follows:

- We demonstrate that static latent compressors are brittle under conversational dynamics, degrading across consecutive compressions and collapsing under turn-by-turn rollout.

- We introduce Context-Driven Incremental Compression (C-DIC), which is, to our knowledge, the first framework for turn-level incremental compression within a single compact dialogue memory; a *retrieve* → *revise* → *write-back* scheme with retrieval-aware TBPTT enables cross-turn sharing and correction, yielding long-range behavior without full-history re-encoding or backpropagation.

- C-DIC improves long-range coherence and reference fidelity while greatly reducing inference cost and input size, outperforming truncation, summarization, and static compression baselines. Notably, C-DIC shows stable inference latency and perplexity over hundreds of dialogue turns.

## 2. Preliminaries & Related Work

### 2.1. Multi-Turn Dialogue Generation

Multi-turn interaction enables conversational agents to sustain coherent, goal-oriented discourse by using prior turns to resolve coreference, track user preferences, and revise assumptions. Such continuity is indispensable in real-world settings, where users expect the system to remember and adapt to prior context.

Formally, let a dialogue with $T$ turn be a sequence $\mathcal{D}_{1:T} = \{(q_1, r_1), \ldots, (q_T, r_T)\}$, where $q_t$ is the user query and $r_t$ the system response at turn $t$. At each turn, a model predicts $r_t$ conditioned on the current query and the previous dialogue history $\mathcal{D}_{<t}$, and training maximizes

$$\log P_\theta(r_t \mid q_t, \mathcal{D}_{<t}). \tag{1}$$

Under full-context prompting, the full history $\mathcal{D}_{<t}$ must be supplied at *every* subsequent turn since each future prediction depends on it. If each exchange contributes on average $L$ tokens, the prompt length at turn $t$ is $tL$, and the cumulative self-attention cost over a $T$-turn dialogue scales as $\sum_{t=1}^{T} O((tL)^2) = O(T^3 L^2)$ (Tay et al., 2022). This cubic growth rapidly dominates latency and memory usage, motivating context management methods that reduce the effective conditioning context. We discuss the role of key-value caching in Appendix A.

### 2.2. Textual Context Management

Text-based approximations of the dialogue history are common strategies for mitigating the inefficiencies of full-context encoding in multi-turn dialogues. The simplest approach is truncation, where only the most recent $k$ utterances from the history are retained (Xu et al., 2022; Laban et al., 2025). While effective in limiting input size, truncation can discard earlier information needed later. To retain more of the dialogue semantics, summarization-based methods compress history into a natural language summary (Wang et al., 2025; Packer et al., 2024), which may omit details or become stale as the conversation evolves. Retrieval and long-context inference methods (Lewis et al., 2021; Xiao et al., 2024) selectively reuse relevant history segments, while prompt compression methods (Jiang et al., 2023) shorten the textual prompt before generation. These methods manage textual context through summaries, retrieval, compression, or virtual memory, rather than learning a latent dialogue state that is retrieved, revised, and written back across turns.

### 2.3. Latent Context Compression

To move beyond text proxies, recent work proposes latent context compression, which maps variable-length context to a fixed set of latent vectors (Wingate et al., 2022; Mu et al., 2024; Chevalier et al., 2023; Ge et al., 2024). A common formulation appends $k$ trainable compression tokens $C \in \mathbb{R}^{k \times d}$ to the input and runs the language model once. The hidden states at those positions are kept as a dense matrix $\mathbf{Z} \in \mathbb{R}^{k \times d}$ for downstream conditioning. This gives the generator a fixed-size latent conditioning interface determined by $k$.

AutoCompressor (Chevalier et al., 2023) recursively *accumulates* compression embeddings over context segments. ICAE (Ge et al., 2024) uses a modular autoencoder architecture in which a compressor maps the input context to a fixed-size latent matrix $\mathbf{Z}$, which is then consumed by a frozen generator. In its standard one-shot setting, $\mathbf{Z}$ is produced once and remains static rather than being revised across turns. While effective for static or one-shot inputs, these compressors are less suited to evolving dialogue: reflecting new information typically requires recompressing the available context or appending new latents without explicit revision, and a fixed latent capacity can increase the risk of forgetting as the dialogue grows.

Memory-augmented long-context models such as Rae et al. (2019); Bulatov et al. (2022); Chevalier et al. (2023) introduce compressed past segments, recurrent memory slots, or accumulated embeddings. However, they primarily define memory at the token or segment level. In contrast, C-DIC treats compression as dialogue-level memory update: latent states are retrieved, revised, and written back across turns. By coupling multi-slot retrieval, incremental latent write-back, and retrieval-aware truncated credit assignment, C-DIC provides revisable dialogue memory without repeatedly re-encoding the full history or modifying the base generator.

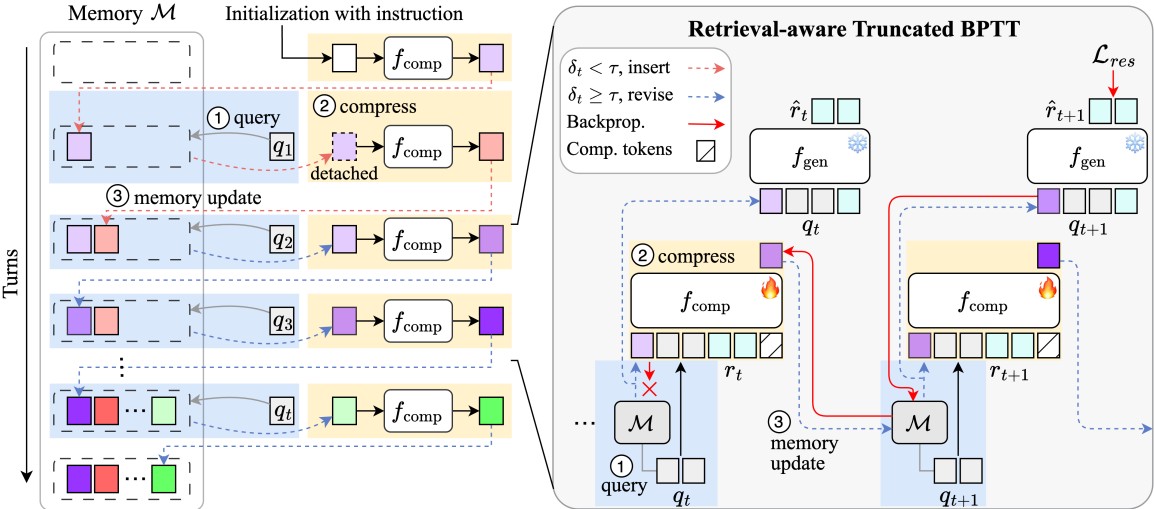

*Figure 3.* **Overview of retrieval-conditioned incremental compression and ra-TBPTT.** *Left*: We maintain memories $\mathcal{M}$ that store a set of compressed thread states that evolve turn by turn. The memories are initialized by compressing the instruction into a thread state. Each turn then follows three steps. **(1) Query**: given $q_t$, we score the existing thread states and retrieve the relevant states; if the best matching thread is topically irrelevant, we still fetch it for continuity but *detach* it. **(2) Compress**: the trainable compressor $f_{\text{comp}}$ summarizes the retrieved states and the current turn into a new *thread state*, detailed on right. **(3) Memory update**: we apply a gradient-free memory update rule using the peak similarity $\delta_t$ (red dashed = *insert* (topic shift), blue dashed = *revise* the best-matching state (on-topic)); see 3.2 for details. *Right*: For the training of $f_{\text{comp}}$, a per-turn response loss $\ell_t$ is minimized, enhanced with a retrieval-aware TBPTT: gradients flow *one hop* along the arg-max usage edge and are truncated ($\to \times$) thereafter. At turn $t$, the compressor $f_{\text{comp}}$ summarizes $(\mathcal{R}_t, q_t, r_t, C)$ into a new state and writes it back; the frozen response generator $f_{\text{gen}}$ conditions on $\mathcal{R}_t$ to produce $\hat{r}_t$.

# 3. Methodology: Context-Driven Incremental Compression

Recall that conventional prompting strategies suffer from inefficiency issues and fail to provide contextually grounded responses. To handle these issues, we propose **Context-Driven Incremental Compression (C-DIC)**, a framework for scalable multi-turn dialogue modeling that enables efficient, context-sensitive reuse of prior interactions, as shown in Figure 3. We model a dialogue as interleaved contextual threads and maintain a compact memory whose slots store revisable, per-thread compressed states. At each turn, the system executes a light retrieve $\to$ compress $\to$ write-back loop that circumvents the repeated encoding of entire history while allowing cross-turn sharing and correction. While we freeze the response generator, we optimize only the compressor and learnable compression tokens during training. We further discuss this training setup and the rationale for this design in Appendix B.

## 3.1. Compressor Initialization

Instead of training a compressor from scratch, we initialize the compressor with a pretrained checkpoint of ICAE (Ge et al., 2024), which was trained on large-scale corpora for one-shot document compression. We adapt this initialized compressor to our incremental, retrieval-conditioned setting

with a frozen response generator. This approach leverages the pretrained compression capability of ICAE, endowing our compressor with high-capacity, context-faithful representations without incurring additional pretraining cost.

## 3.2. Incremental Compression & Context-Aware Retrieval

Our design targets three needs: efficiency over long histories, coherence within an active thread, and concentrated supervision only on the retrieved memory states. Instead of re-encoding the full, ever-growing history at every turn, we maintain a compact memory $\mathcal{M}_{<t} = \{\mathbf{Z}_i\}$ of compressed thread states that evolve with ongoing dialogue. At turn $t$, we (i) **retrieve** a small, query-related subset $\mathcal{R}_t \subset \mathcal{M}_{<t}$ for conditioning, (ii) **generate** the response with a frozen decoder, and (iii) **compress** the new turn into an updated memory slot via a gradient-free write-back policy.

**Turn-wise compression (base case).** We first describe compression without retrieval to fix notation and the learning signal. Given the input pair $(q_t, r_t)$, the compressor produces a compact summary

$$\mathbf{Z}_t = f_{\text{comp}}([\text{Emb}(q_t); \text{Emb}(r_t); \mathbf{C}]; \theta), \quad (2)$$

where $q_t$ and $r_t$ are the turn-$t$ query and gold response, $\mathbf{C} \in \mathbb{R}^{n \times d}$ is embedding of learnable compression tokens,

$\theta$ are the compressor parameters, and $\mathbf{Z}_t \in \mathbb{R}^{n \times d}$ is the resulting compressed state. The state $\mathbf{Z}_t$ is written to memory and becomes available to condition later responses; the generator's parameters remain fixed, so learning focuses on producing compressed contexts that are useful when consulted. This base case already yields a compact latent conditioning interface and a well-defined supervision signal, but it treats all prior context symmetrically and cannot adapt granularity to what the model actually reuses.

**From turn-wise to retrieval-based thread compression**
To make compression conditional on the context that actually matters at turn $t$, we introduce a retrieved support set $\mathcal{R}_t \subset \mathcal{M}_{<t}$. Each slot $\mathbf{Z}_i$ is scored by a semantic match with mild recency decay:

$$S(q_t, \mathbf{Z}_i) = \frac{\langle \psi(f_{\mathrm{comp}}(q_t, C)),\ \psi(\mathbf{Z}_i)\rangle}{\|\psi(f_{\mathrm{comp}}(q_t, C))\|\ \|\psi(\mathbf{Z}_i)\|}\ e^{-\alpha \Delta t_i}, \quad (3)$$
$$\mathcal{R}_t = \{\mathbf{Z}_i :\ S(q_t, \mathbf{Z}_i) > \tau\}.$$

Here $\psi(\cdot)$ is a pooling function (e.g. mean or CLS token) over token-level representations, $\Delta t_i$ is the number of turns since $\mathbf{Z}_i$ was last retrieved, and $\alpha$ is decay rate, and $\tau$ is a fixed retrieval threshold on the similarity score $S$. If no slot exceeds $\tau$, we fall back to the single best match $\{\mathbf{Z}_{\arg\max_i S(q_t, \mathbf{Z}_i)}\}$. The response generator conditions on the retrieved supports rather than the entire history:

$$\hat{r}_t = f_{\mathrm{gen}}([\mathcal{R}_t; \mathrm{Emb}(q_t)]; \phi). \quad (4)$$

Crucially, compression becomes *incremental* with respect to these supports:

$$\mathbf{Z}_t = f_{\mathrm{comp}}([\mathcal{R}_t; \mathrm{Emb}(q_t); \mathrm{Emb}(r_t); C]; \theta). \quad (5)$$

This retrieval conditioning focuses $\mathbf{Z}_t$ on the **active thread**, improving long-horizon coherence while keeping per-turn computation proportional to $|\mathcal{R}_t|$ rather than the dialogue length.

**Write-back and thread continuity** To keep the memory both compact and faithful to the evolving topic, we define a *deterministic, gradient-free* update rule. At turn $t$, score the current query against existing slots

$$\delta_t = \max_i S(q_t, \mathbf{Z}_i), \qquad j_t = \arg\max_i S(q_t, \mathbf{Z}_i),$$

and update the memory by either inserting a new state (topic shift) or revising the most similar state (thread continuation):

$$\mathcal{M}_{<t+1} = \begin{cases} \mathcal{M}_{<t} \cup \{\mathbf{Z}_t\}, & \text{if } \delta_t < \tau, \\ (\mathcal{M}_{<t} \setminus \{\mathbf{Z}_j\}) \cup \{\mathbf{Z}_t\}, & \text{otherwise.} \end{cases} \quad (6)$$

Here $\tau$ is the retrieval threshold used in Section 3.2. This policy preserves *thread continuity* by updating the best-matching slot when relevant, and by opening a new slot

---

**Algorithm 1** Inference: Retrieve → Generate → Compress → WriteBack

---

**Input:** compressor $f_{\mathrm{comp}}$; frozen generator $f_{\mathrm{gen}}$; tokens $C$; threshold $\tau$; decay $\alpha$
**Output:** generated responses $\{\hat{r}_t\}_{t=1}^T$
$\mathcal{M} \leftarrow \varnothing$
**for** $t = 1$ **to** $T$ **do**
  **if** $\mathcal{M} \neq \varnothing$ **then**
    $s_i \leftarrow S(q_t, \mathbf{Z}_i)$ for all $\mathbf{Z}_i \in \mathcal{M}$
    $j \leftarrow \arg\max_i s_i$
  **end if**
  **if** $\mathcal{M} \neq \varnothing$ **and** $\max_i s_i \geq \tau$ **then**
    $\mathcal{R}_t \leftarrow \{\mathbf{Z}_i : s_i \geq \tau\}$     // (1) Multi-slot retrieval
  **else if** $\mathcal{M} \neq \varnothing$ **then**
    $\mathcal{R}_t \leftarrow \{\mathbf{Z}_j\}$     // Top-1 fallback for generation
  **else**
    $\mathcal{R}_t \leftarrow \varnothing$
  **end if**
  $\hat{r}_t \leftarrow f_{\mathrm{gen}}([\mathcal{R}_t; q_t])$
  $\mathbf{Z}_t \leftarrow f_{\mathrm{comp}}(\mathcal{R}_t, q_t, \hat{r}_t, C)$   // (2) Generate & compress
  **if** $\mathcal{M} = \varnothing$ **then**
    $\mathcal{M} \leftarrow \{\mathbf{Z}_t\}$
  **else if** $\max_i s_i < \tau$ **then**
    $\mathcal{M} \leftarrow \mathcal{M} \cup \{\mathbf{Z}_t\}$     // Insert new state
  **else**
    $\mathcal{M} \leftarrow (\mathcal{M} \setminus \{\mathbf{Z}_j\}) \cup \{\mathbf{Z}_t\}$   // (3) Memory update
  **end if**
  Update recency counters $\Delta t_i$ for all $\mathbf{Z}_i \in \mathcal{M}$
**end for**

---

when relevance is low. Because no gradients flow through selection or write-back, inference remains lightweight and avoids dependence on the raw dialogue-token history. We provide a more detailed discussion of alternative memory update variants in Appendix C.

**Retrieval-aware truncated BPTT** In multi-turn dialogue with compressed memory, the model does not attend to the full history; it consults a tiny set of states selected by retrieval. Standard BPTT (Werbos, 1990) backpropagates through *all* past turns (costly and misaligned with usage), while conventional TBPTT (Schmidt, 2019) truncates by a fixed window (agnostic to which turns were actually consulted). We therefore use a *retrieval-aware* truncated computation graph that assigns credit only along the selected memory-update path.

Specifically, we minimize the per-turn negative log-likelihood

$$\mathcal{L} = \frac{1}{T} \sum_{t=1}^{T} \ell_t, \quad (7)$$
$$\ell_t = -\log P_\phi(r_t \mid q_t, \mathcal{R}_t),$$

where $r_t$ is the gold response used for teacher-forced training. We perform a reverse-time backward pass with a *one-hop* truncation along the memory updated chain:

$$\frac{\partial \ell_t}{\partial \mathbf{Z}_{j_t}} \neq 0 \quad \text{iff} \quad \delta_t \geq \tau,$$
$$\frac{\partial \ell_t}{\partial \mathbf{Z}_s} = 0 \quad \text{for all } s \neq j_t. \tag{8}$$

Equivalently, with a mask $M_{s,t} = \mathbb{1}[s = j_t] \cdot \mathbb{1}[\delta_t \geq \tau]$,

$$\frac{\partial \mathcal{L}}{\partial \mathbf{Z}_s} = \sum_{t=1}^{T} M_{s,t} \frac{\partial \ell_t}{\partial \mathbf{Z}_s}.$$

Thus, retrieved states that are not selected for write-back are used as stop-gradient context. For off-topic turns ($\delta_t < \tau$) we keep the arg-max slot for *forward* continuity but detach it in training:

$$\tilde{\mathbf{Z}}_{j_t} = \text{stopgrad}(\mathbf{Z}_{j_t}), \qquad \delta_t < \tau \Rightarrow \frac{\partial \ell_t}{\partial \mathbf{Z}_{j_t}} = 0,$$

so credit never flows into mismatched memory. This objective provides sparse, retrieval-aware credit assignment for the memory path used by C-DIC's write-back operation. Importantly, Eq. 8 describes the implemented truncated computation graph, not full backpropagation over all retrieved states or all possible retrieval paths. We further discuss limitations of full BPTT and fixed-window TBPTT in Appendix D.

**Inference** Algorithm 1 outlines the inference procedure. At each turn, C-DIC first scores all memory states against the current query and retrieves the relevant subset for generation. The frozen generator then produces $\hat{r}_t$ conditioned on the retrieved latent states and the user query. We then compress the turn into a new thread state and update memory (insert on topic shift, otherwise revise the active state); inference is fully gradient-free, keeping latency low.

## 4. Experiments

### 4.1. Datasets

To measure long-horizon coherence, we follow the setting of existing works (Xu et al., 2022) and evaluate on Multi-Session Chat (MSC) (Xu et al., 2022) and REALTALK (Lee et al., 2025), two recent multi-session corpora structured around re-engagements occurring after hours or days. All datasets used are publicly available for research purposes.

MSC contains human-human conversations spanning up to five sessions. We use the official training split with 1 001 episodes (averaging 53.3 utterances). For evaluation, we leverage sessions 2–5, yielding an average of 66 utterances per conversation.

REALTALK is a real-world WhatsApp-style corpus featuring 10 conversations collected across 21 days, averaging 21.9 sessions and 894.4 utterances per conversation. To effectively validate the robustness and transferability to longer context, we evaluate on REALTALK in a zero-shot setting. Also, we use two evaluation settings for REALTALK: *all-sessions*, which includes cross-session history to test long-term context, and *per-session*, which restricts inputs to the current session only.

To assess whether MSC and REALTALK require long-range context and whether target responses are generic, we provide an LLM-based dataset characterization with human verification in Appendix E. We construct MSC-QA using GPT-4o, generating QA pairs whose answers depend on earlier contextual mentions in MSC dialogues, and use it as a targeted diagnostic for earlier-context use. Finally, we report LongMemEval (Wu et al., 2025) as an auxiliary long-context QA diagnostic beyond the main dialogue-generation benchmarks.

### 4.2. Baselines

We compare against raw-context prompting, textual retrieval, prompt-compression, and latent-compression baselines under the same evaluation protocol; implementation details are provided in Appendix B.

**Raw-context baselines.** **Full prompting** uses the available history up to the context limit, **Full prompting[§]** is a stronger-backbone reference using `Llama3.1-8B` with a longer context window, and **Truncation** keeps only the last $k=5$ turns.

**Textual retrieval and prompt-compression baselines.** **Summarization** recursively summarizes dialogue history with the shared frozen `Llama-2-Chat-7B`. **RAG@K** retrieves the top-$K$ prior turns from the same dialogue by semantic similarity and concatenates them with the current query for response generation; we evaluate $K \in \{5, 10, 20\}$. **RAG-threshold** retrieves prior turns whose similarity to the current query exceeds a tuned threshold. **LLMLingua** (Jiang et al., 2023) compresses the textual prompt before generation, while **InfLLM** (Xiao et al., 2024) provides a training-free long-context inference baseline that retrieves relevant context blocks at generation time.

**Latent-compression baselines.** **AutoCompressor** (Chevalier et al., 2023) recursively compresses history chunks into learned compression tokens, accumulating compressed context. **ICAE** (Ge et al., 2024) uses an autoencoder-style compressor with a frozen generator; we evaluate three variants: **incremental**, which relays previous latents through each update, **one-shot**, which re-encodes the full available context at each turn (original setting), and

*Table 1.* **Main results on MSC and REALTALK.** We report perplexity (PPL), BLEU, and ROUGE (R-L, R-1, R-2), where REALTALK results are zero-shot. On REALTALK, we use the *per-session* setting due to GPU memory limits of several compression baselines, and text-only baselines are truncated to their maximum context length. Best results are in bold, and second-best results are underlined among methods using the shared 7B backbone. The [§] row is a stronger-backbone full-prompting reference using Llama3.1-8B with a longer context window. Overall, these results highlight the benefit of revisable latent memory for long-form dialogue modeling under fixed backbone and memory constraints.

| Models | MSC | | | | | REALTALK | | | | |
|---|---|---|---|---|---|---|---|---|---|---|
| | PPL↓ | BLEU↑ | R-L↑ | R-1↑ | R-2↑ | PPL↓ | BLEU↑ | R-L↑ | R-1↑ | R-2↑ |
| Full prompting | 41.245 | 0.008 | 0.110 | 0.157 | 0.015 | 25.546 | 0.022 | 0.110 | 0.160 | 0.020 |
| Full prompting[§] | 24.244 | 0.015 | 0.116 | 0.157 | 0.017 | 29.444 | 0.017 | 0.109 | 0.158 | 0.018 |
| Truncation | 30.890 | 0.012 | 0.128 | 0.184 | 0.024 | 23.830 | 0.023 | 0.114 | 0.165 | 0.022 |
| Summarization | 41.849 | 0.013 | 0.128 | 0.172 | 0.024 | 26.087 | 0.023 | 0.114 | 0.168 | 0.023 |
| RAG@5 | 35.530 | 0.008 | 0.110 | 0.148 | 0.014 | 26.789 | 0.020 | 0.103 | 0.151 | 0.015 |
| RAG@10 | 34.630 | 0.011 | 0.110 | 0.157 | 0.014 | 27.311 | 0.019 | 0.102 | 0.148 | 0.014 |
| RAG@20 | 31.179 | 0.013 | 0.111 | 0.156 | 0.014 | 27.646 | 0.019 | 0.102 | 0.148 | 0.014 |
| RAG-threshold | 40.057 | 0.011 | 0.109 | 0.157 | 0.013 | 28.238 | 0.009 | 0.101 | 0.147 | 0.013 |
| LLMLingua | 36.211 | 0.012 | 0.105 | 0.157 | 0.011 | 31.067 | 0.008 | 0.092 | 0.131 | 0.011 |
| InfLLM | 27.329 | 0.016 | 0.118 | 0.161 | 0.019 | 29.995 | 0.013 | 0.099 | 0.141 | 0.014 |
| AutoCompressor | 9.285 | 0.012 | 0.121 | 0.145 | 0.019 | 12.625 | 0.019 | 0.055 | 0.134 | 0.019 |
| ICAE (incremental) | 513.774 | 0.006 | 0.057 | 0.069 | 0.005 | 124.024 | 0.020 | 0.068 | 0.086 | 0.013 |
| ICAE (one-shot) | 27.656 | 0.017 | 0.133 | 0.190 | 0.027 | 21.390 | 0.025 | 0.118 | 0.166 | 0.026 |
| ICAE (append) | *Out of memory* | | | | | *Out of memory* | | | | |
| Ours | **8.431** | **0.023** | **0.160** | **0.205** | **0.037** | **9.789** | **0.035** | **0.134** | **0.176** | **0.030** |

**append**, which concatenates newly compressed latents without revision (growing latent length).

All methods use the shared frozen `Llama-2-Chat-7B` (Touvron et al., 2023) generator, except methods that require their own fine-tuning setup and the [§] stronger-backbone reference. For fairness, all learned compression baselines are adapted on MSC for 2 epochs and evaluated zero-shot on REALTALK without additional tuning. This keeps the main comparison focused on *context management* rather than decoder capacity or dataset-specific tuning.

### 4.3. Evaluation

We adopt a set of widely used, complementary metrics aligned with common practices in dialogue and summarization research. Following prior works (Chevalier et al., 2023; Ge et al., 2024), we employ perplexity (PPL) to measure the generative fluency. For measuring content alignment with human responses, we report BLEU (Papineni et al., 2002) (up to 4-gram precision) and ROUGE (Lin, 2004) as standard reference-based metrics. BLEU computes n-gram precision against reference responses. R-1 (ROUGE-1) and R-2 (ROUGE-2) measure unigram and bigram recall, indicating lexical coverage, while R-L (ROUGE-L) utilizes the longest common subsequence to reflect structural similarity and fluency. For MSC-QA, we additionally report answer accuracy using the LongMemEval (Wu et al., 2025) evaluation protocol: GPT-4o judges a prediction as correct if it

contains the required answer implied by the reference.

### 4.4. Results

Table 1 presents a comprehensive evaluation of our framework on the MSC and REALTALK datasets. C-DIC achieves the best results across PPL, BLEU, and ROUGE in our evaluations, while improving efficiency by conditioning generation on a compact retrieved latent context rather than the full dialogue history.

Unlike full-prompting, truncation, or summarization baselines that process raw text linearly, our approach incrementally compresses dialogue turns into fixed-size latent representations and retrieves only memories relevant to the current query. This retrieval-aware design keeps the effective conditioning signal focused and prevents quality from deteriorating as dialogues grow longer, yielding strong fluency and coherence in long-horizon settings.

Compared to prior latent compression approaches such as AutoCompressor and ICAE, which are designed for static, one-shot settings, our model achieves superior performance with similar or lower memory usage. This demonstrates the effectiveness of our incremental compression strategy and retrieval-based memory refinement in the evolving dialogue. In particular, the naive incremental ICAE baseline catastrophically fails (PPL $\approx 513$). This is due to a structural mismatch between ICAE's one-shot training objective and repeated compression, which we discuss in Appendix F.

*Table 2.* **Closed-loop vs. teacher-forcing on the REALTALK (all-sessions).** Results for our method under teacher forcing (ground-truth history) and closed-loop generation (conditioning on the model's own past responses) over the full multi-session history.

| Setting | PPL↓ | BLEU↑ | R-L↑ | R-1↑ | R-2↑ |
|---|---|---|---|---|---|
| Teacher-forcing | **9.556** | **0.036** | **0.140** | **0.184** | **0.035** |
| Closed-loop | 9.576 | **0.036** | 0.137 | 0.182 | 0.032 |

*Table 3.* **MSC-QA results.** Targeted QA evaluation for probing long-range dialogue context use, reported with accuracy (Acc.) and standard generation metrics.

| Models | Acc.↑ | PPL↓ | BLEU↑ | R-L↑ | R-1↑ | R-2↑ |
|---|---|---|---|---|---|---|
| Full Prompting | 0.042 | 30.560 | 0.016 | 0.121 | 0.160 | 0.027 |
| ICAE (incremental) | 0.000 | 821.209 | 0.009 | 0.057 | 0.066 | 0.004 |
| ICAE (one-shot) | 0.004 | 38.268 | 0.010 | 0.095 | 0.125 | 0.012 |
| InfLLM | 0.062 | 8.963 | 0.020 | 0.146 | 0.181 | 0.040 |
| Ours | **0.104** | **5.950** | **0.039** | **0.205** | **0.243** | **0.064** |

Finally, Table 2 reports our REALTALK *all-sessions* results under both teacher forcing and closed-loop generation; most baselines cannot be evaluated in this setting without truncation due to GPU memory limits. Notably, our model generalizes robustly to REALTALK, a much longer, more open-domain dataset, despite being trained only on MSC. As a zero-shot evaluation, this result underscores our method's adaptability on both domain and length shifts. Moreover, the small difference between teacher forcing and closed-loop suggests stable long-horizon behavior over hundreds of turns. We further report mean±std over three random seeds with seed-level significance tests in Appendix G, and additional closed-loop comparisons on REALTALK in Appendix H.

### 4.5. Long-Range Context Diagnostics

Standard generation metrics such as PPL, BLEU, and ROUGE are useful for evaluating next-response quality, but they are indirect measures of long-range dialogue fidelity. We therefore complement the main generation results with targeted diagnostics that directly probe earlier-context use.

First, we evaluate MSC-QA, a question-answering diagnostic constructed from MSC dialogues to test whether models can recover information from earlier turns. As shown in Table 3, C-DIC achieves the highest accuracy among the evaluated methods while also improving PPL, BLEU, and ROUGE. This provides complementary evidence that C-DIC's gains are associated with better use of compressed long-range dialogue context, rather than improvements in lexical overlap alone.

Beyond MSC-QA, we report REALTALK *all-sessions* closed-loop results in Table 2, dataset-level evidence that many target responses require earlier context in Appendix E,

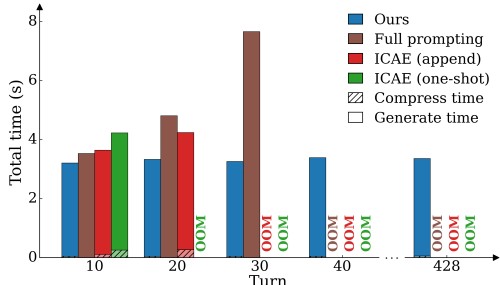

*Figure 4.* **Latency vs. dialogue length.** Total wall-clock time (s) to process a single dialogue when the maximum number of turns is capped at {10, 20, 30, 40, 428}. Bars compare **Ours**, Full prompting, ICAE (append), and ICAE (one-shot). The hatched segment denotes *compression time* and the solid segment denotes *generation* time; "OOM" denotes exceeding the memory budget under the corresponding turn cap. Evaluations use REALTALK in the *all-sessions* setting by truncating to the most recent turns.

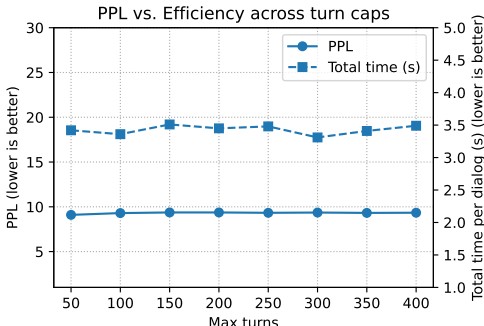

*Figure 5.* **Latency vs. performance across turn caps.** Dual–axis plot with *PPL* (left) and *total time per dialog* in seconds (right) versus the maximum number of retained turns (50-400). Evaluations use the REALTALK all-sessions setting with histories truncated to the most recent turns.

and LongMemEval QA results with explicit correctness targets in Appendix M. Together, these diagnostics evaluate long-range context use beyond standard lexical generation metrics and provide strong converging evidence that C-DIC's improvements are driven by superior long-range context utilization.

### 4.6. Latency Comparison

Figure 4 and 5 report total wall-clock time per dialogue as we cap the maximum number of retained turns at {10, 20, 30, 40, 428}. Under this evaluation setting, C-DIC maintains stable empirical latency of approximately ∼ 3–3.5s across turn caps, with compression accounting for only a small fraction of the total runtime. In contrast, Full prompting grows rapidly (≈ 3.6s@10, ≈ 4.7s@20, ≈ 7.7s@30) and becomes OOM beyond 30 turns. ICAE (append) is slower than ours at 10–20 turns and becomes OOM from 30 onward; ICAE (one-shot) is slower at 10 and

*Table 4.* **Ablation study on the REALTALK dataset.** We evaluate the contribution of incremental compression (IC), retrieval-aware truncated backpropagation through time (R-TBPTT) and memory-based context threading (MCT) by removing each component from the full model. All variants are evaluated at the final turn of each conversation under the *all-sessions* setting to assess long-term generation quality.

| Models | PPL↓ | BLEU↑ | R-L↑ | R-1↑ | R-2↑ |
|---|---|---|---|---|---|
| C-DIC | 9.356 | **0.069** | **0.173** | **0.213** | **0.056** |
| (−) IC | 25.527 | 0.040 | 0.075 | 0.103 | 0.018 |
| (−) R-TBPTT | 12.295 | 0.025 | 0.119 | 0.172 | 0.018 |
| (−) MCT | **9.197** | 0.046 | 0.128 | 0.188 | 0.025 |

becomes OOM already at 20. At 30 turns, C-DIC is roughly $2.4\times$ faster than Full prompting.

The key advantage of C-DIC is that generation conditions on a small set of retrieved latent states rather than repeatedly exposing the generator to an ever-growing raw dialogue history. Within the same hardware and context-length setting, *C-DIC is the only evaluated method that handles 428 turns*. This demonstrates that retrieval-conditioned incremental compression provides a substantially more scalable generation path than full-history prompting or static latent compression in long-dialogue settings. For detailed latency components, see Table 12 in Appendix I. Note that OOM is setting-dependent: ICAE(append) scales primarily with the *number of turns*, whereas ICAE(one-shot) is constrained mainly by *total context length*. Consequently, their OOM thresholds can differ between the *all-sessions* latency evaluation here and the *per-session* results in Table 1.

C-DIC does not require the total memory bank to be formally bounded itself: stored memory slots may grow with dialogue length. However, the retrieved subset used for generation remains small in practice. We also evaluate a bounded retrieval variant that caps the number of retrieved slots, which keeps the generator's latent context size fixed while achieving comparable performance in our experiments. We provide memory-growth, bounded-retrieval, and retrieval-overhead analyses in Appendix J.

### 4.7. Ablations

We ablate three components of our system: *incremental compression* (IC), *retrieval-aware truncated BPTT* (R-TBPTT), and the *memory module* (retrieval + write-back) for context threading. As shown in Table 4, removing IC causes the greatest degradation. PPL rises from **9.356** to **25.527** and ROUGE-2 drops from **0.056** to **0.018**, indicating that turn-wise compression and revision are critical for preserving salient content. Disabling R-TBPTT also degrades supervision quality, confirming the benefit of backpropagating one hop along the actual retrieval path. Although removing memory-based context threading gives slightly better

PPL, it yields markedly worse BLEU and recall (ROUGE scores), implying our proposed context threading greatly improves the long-horizon coherence. In general, the full model (C-DIC) achieves best performance among compared baselines under a shared 7B and demonstrates the necessity and complementary gains of all IC, R-TBPTT, and memory-based context threading. Additional sensitivity analyses for $\tau$ and $\alpha$, including memory-growth effects, are provided in Appendix K.

## 5. Limitations

C-DIC has several limitations. First, the total number of stored memory slots can grow when dialogue repeatedly shifts topics, although generation uses only a small retrieved subset and bounded retrieval can fix the generation-side context size. Second, C-DIC currently relies on pretrained latent-compression components, so results may vary with different compressor initializations or backbone models. Third, persistent latent memories can retain sensitive information and may propagate stale or incorrect content across later turns; latent compression should therefore not be viewed as a privacy or deletion mechanism. Finally, our evaluation remains limited to a finite set of benchmarks and diagnostics; broader domains and evaluation protocols remain important directions for future work.

## 6. Conclusion

In this paper, we present **Context-Driven Incremental Compression (C-DIC)**, a thread-aware dialogue memory for long conversations. It replaces full-context prompting with a lightweight *retrieve → revise → write-back* loop trained via retrieval-aware truncated BPTT, enabling cross-turn context sharing and revision without re-encoding the entire history. By modeling conversations as interleaved threads and conditioning generation on a compact set of retrieved latent states, C-DIC provides a practical mechanism for using long-range context under limited context and compute budgets. Empirically, it remains stable where static compressors collapse under multi-turn rollout, outperforming truncation, summarization, and static latent baselines while reducing inference costs. In our latency evaluation, C-DIC maintains stable empirical end-to-end latency of approximately 3∼3.5s despite growing dialogue history, and is the only method demonstrated to handle up to 428 turns under the same hardware, underscoring the scalability of our incremental, retrieval-conditioned design. Ablations confirm that each component—incremental compression, retrieval-aware TBPTT, and memory-based context threading—contributes materially to the overall gains in coherence and faithfulness.

## Impact Statement

This work improves the efficiency and robustness of multi-turn dialogue generation by introducing a compact, thread-aware dialogue memory that is updated incrementally over the course of a conversation. By avoiding repeated computation over an ever-growing dialogue history, the proposed approach can reduce inference-time latency and memory overhead, thereby making long-horizon conversational systems more scalable and practical to deploy. As with other retrieval- and memory-based approaches, reusing prior context can carry forward undesirable artifacts from earlier turns; however, our method introduces no new risk categories beyond this established setting. Although compressed representations avoid storing verbatim logs, they may still retain user-specific information; deployments should pair them with retention/deletion policies and privacy-aware filtering.

## Acknowledgements

Lei Chen is supported by National Key Research and Development Program of China Grant No. 2023YFF0725100, National Science Foundation of China under Grant No. U22B2060, Guangdong-Hong Kong Technology Innovation Joint Funding Scheme Project No. 2024A0505040012, AOE Project AoE/E-603/18, Theme-based project TRS T41-603/20R, CRF Project C2004-21G, Key Areas Special Project of Guangdong Provincial Universities 2024ZDZX1006, Guangdong Province Science and Technology Plan Project 2023A0505030011, HKUST(GZ) CMCC(Guangzhou Branch) Metaverse Joint Innovation Lab under Grant No. P00659, Hong Kong ITC TC-SKLCRCC26EG01, ITF grant PRP/004/22FX, Zhujiang scholar program 2021JC02X170, HKUST Webank joint research lab. Jiachuan Wang is supported in part by JST CREST (JPMJCR22M2). Yongqi Zhang's work is supported by Guangdong Provincial Natural Science Foundation 2025A1515010304, Guangdong Province Project 2024QN11X088, Guangzhou Science and Technology Planning Project 2025A03J4491. This work was supported in part by compute resources provided by NVIDIA.

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

## A. Key-Value Caching

A standard engineering mitigation for inference is *key–value (KV) caching*, which stores the hidden states of past tokens layer-by-layer so that only the newest tokens are processed afresh (Tay et al., 2022). Caching indeed reduces *compute* for *unchanged* prefixes, but it introduces several limitations that are critical in an interactive setting. First, any user edit invalidates the cache from the edit point onward, which forces a full recomputation of attention. Second, KV caching trades FLOPs for memory: the cache footprint grows linearly with both dialogue length and layer count, quickly exhausting GPU memory when thousands of sessions run concurrently. Third, caching leaves the *attention distribution* unchanged, so the model still under-attends to mid-history tokens, a known positional-bias issue that harms long-range coherence.

## B. Implementation Details

We implement our models with the `Llama2-Chat-7B` backbone by adapting the ICAE (Ge et al., 2024) checkpoint to the multi-turn dialogue setting. This provides a strong initialization for compression and allows us to focus on the effect of incremental, thread-aware memory. For practicality and to isolate the contribution of the memory mechanism, we freeze all weights of the base model and update only the LoRA-adapted compressor and the learnable compression tokens during training. All training and inference were conducted using an NVIDIA A100 80GB GPU. Fine-tuning required around 17 GPU hours on a single A100 GPU. Across all experiments employing the ICAE checkpoint, we use a compression token length of 128, a retrieval threshold $\tau = 0.8$, and cosine similarity with exponential decay (decay rate $\alpha = 0.05$. For training, we use a batch size of 1 while inference is performed with batch sizes of 8 and 1 for the MSC and REALTALK datasets, respectively. We finetune our model for 2 epochs, using AdamW with a learning rate of $2 \times 10^{-4}$.

## C. Alternative Memory Update Strategies

We implemented two alternatives to the simple replacement-based write-back: (i) Exponential Moving Average (EMA) updates with decay factors $\beta \in \{0.3, 0.5, 0.7\}$, using $\beta \cdot \text{old\_memory} + (1 - \beta) \cdot \text{new\_memory}$, and (ii) a 2-layer gating network that learns to interpolate between the old and new memory states. As shown in Table 5, EMA brings at best marginal gains only on the R-1 metric and often yields noticeably worse performance on other metrics, while the gated variant provides only small improvements on some ROUGE scores at the cost of additional complexity. Given this trade-off, we adopt the replacement policy as the simpler and more robust choice.

*Table 5.* **Comparison of memory update strategies on MSC-session 5.** Performance as a function of the selection threshold $\tau$. We report PPL↓, BLEU↑, ROUGE-L↑, ROUGE-1↑, and ROUGE-2↑.

| Strategy | PPL↓ | BLEU↑ | R-L↑ | R-1↑ | R-2↑ |
|---|---|---|---|---|---|
| Replacement | **8.427** | **0.030** | 0.160 | 0.206 | **0.040** |
| EMA ($\beta = 0.3$) | 8.442 | 0.027 | 0.157 | 0.208 | 0.035 |
| EMA ($\beta = 0.5$) | 8.836 | 0.027 | 0.155 | 0.203 | 0.036 |
| EMA ($\beta = 0.7$) | 9.929 | 0.021 | 0.145 | 0.186 | 0.030 |
| Gate | 8.503 | 0.026 | **0.162** | **0.209** | 0.037 |

## D. Limitations of Full BPTT and Fixed-Window TBPTT for Dialogue Memory

Under full BPTT, the gradient with respect to a memory slot $Z_s$ aggregates contributions from all future turns where that slot is actually consulted ($Z_s \in R_t$):

$$\frac{\partial L}{\partial Z_s} = \sum_{t=1}^{T} \mathbf{1}[Z_s \in R_t] \frac{\partial \ell_t}{\partial Z_s}. \tag{9}$$

However, implementing full BPTT requires keeping the computation graph for all $T$ turns in memory, so the activation cost grows linearly with dialogue length. For long conversations this becomes prohibitive in practice and quickly leads to out-of-memory (OOM) errors.

In standard fixed-window TBPTT with horizon $K$, at each turn $t$ all slots older than $K$ steps are detached from the

computation graph before retrieval. Concretely, retrieval reads

$$\tilde{Z}_s = \begin{cases} \text{stopgrad}(Z_s), & s \le t - K, \\ Z_s, & s > t - K, \end{cases} \qquad R_t \subset \{\tilde{Z}_s\}_{s<t}. \tag{10}$$

By the chain rule, for any $s \le t - K$,

$$\frac{\partial \ell_t}{\partial Z_s} = \frac{\partial \ell_t}{\partial \tilde{Z}_s} \frac{\partial \tilde{Z}_s}{\partial Z_s} = \frac{\partial \ell_t}{\partial \tilde{Z}_s} \cdot 0 = 0, \tag{11}$$

even if $Z_s \in R_t$ (i.e., the slot is selected and used at turn $t$). The truncated gradient therefore becomes

$$\frac{\partial L}{\partial Z_s}\bigg|_{\text{TBPTT}} = \sum_{t=1}^{T} \mathbf{1}[Z_s \in R_t]\, \mathbf{1}[t - s < K]\, \frac{\partial \ell_t}{\partial Z_s}, \tag{12}$$

so any selected memory state $Z_s$ that is retrieved only after it falls outside the $K$-step window receives no gradient signal from those distant uses.

## E. Dataset Characterization and Annotation Reliability

We perform additional analysis to quantify (i) how often reference responses depend on distant context in MSC/REALTALK and (ii) how frequently target responses are generic, along with human verification of the LLM judge.

### E.1. Do MSC / REALTALK Require Distant Context?

**LLM-based annotation.** We use GPT-4o to label whether a *candidate past utterance* contains **necessary or materially helpful** information for producing the **reference assistant response** to a target query. To avoid degraded judge reliability on very long prompts, we adopt a pairwise protocol: each instance consists of (i) one historical utterance and (ii) the final-turn context (latest user query + reference response), and the model outputs a binary label (`helpful` / `not helpful`).[1] We then aggregate utterance-level labels into **conversation-level** statistics (e.g., whether any supporting utterance occurs $\ge 10$ turns back).

**LLM-based annotation (genericity).** Separately, we label each reference response as **generic** vs. **not-generic**.[2] A response is *generic* if it can be plausibly reused across many different queries with minimal editing; otherwise it is *not-generic*.

**Sampling.** We sample 500 and 320 conversations from MSC and REALTALK, respectively, restricting to dialogues with $\ge 11$ turns so that "$\ge 10$ turns back" is well-defined. These sample sizes provide stable estimation of conversation-level rates at reasonable cost (worst-case 95% margin $\approx \pm 4$–$6$ percentage points), consistent with cost-aware yet reliable LLM annotation practice (Pangakis et al., 2023).

**Results.** As shown in Table 6, long-range dependencies are common in both datasets under these measures, while generic targets are rare. This suggests that strong performance on MSC/REALTALK is unlikely to be explained solely by short-range cues or templated responses.

### E.2. Human Verification of Judge Reliability

To validate the GPT-4o labels, we run a human verification study with three annotators on 50 randomly sampled items per task and dataset, following the recommended LLM-as-a-judge verification setting in (Calderon et al., 2025). We provide the human verification guidelines and summarize the verification results below.

---

[1]**Instruction 1 (utterance relevance).** You are given the latest user query and the assistant's response of a conversation along with an utterance from the past conversation. Your task is to determine if the utterance is helpful to generating the assistant's response to the latest user query. If it is helpful, respond with `helpful`; otherwise, respond with `not helpful`.

[2]**Instruction 2 (generic response).** You are given the latest user query and the assistant's final response of a conversation. Decide whether the assistant's final response is generic. A response is `generic` if it could be pasted into many different conversations/questions with minimal editing (e.g., greetings/farewells/small talk). Otherwise `not generic`. Output only generic or not generic.

*Table 6.* Dataset characterization via GPT-4o annotation. $n$ = sampled dialogues. Evid. $\geq$10: fraction of *supporting utterances* occurring $\geq$10 turns before the final response. Farthest $\geq$10: fraction of dialogues whose most distant supporting utterance is $\geq$10 turns back. Generic: fraction of target responses labeled generic.

| Dataset | n | Evid. $\geq$**10** (%) | Farthest $\geq$**10** (%) | Generic (%) |
|---|---|---|---|---|
| REALTALK | 320 | 66.94 | 40.31 | 6.25 |
| MSC | 500 | 44.92 | 70.80 | 2.00 |

### E.2.1. HUMAN VERIFICATION GUIDELINES

**A. Helpful-turn verification (utterance-level).**   You are shown: (1) the latest user query, (2) the final assistant reference response, (3) one past utterance from the same dialogue, and (4) the LLM label: `helpful` / `not helpful`.

- **`helpful`**: the past utterance contains information that is *necessary or clearly useful* to produce the final assistant response to the latest query (e.g., key facts, entities, constraints, preferences, or clarifications that the response depends on).

- **`not helpful`**: removing the past utterance would not materially change a reasonable final response (e.g., irrelevant details, small talk).

Mark **Correct = 1** if the LLM's `helpful`/`not helpful` label matches your judgment under the above definition; else **Correct = 0**.

**B. Generic-response verification (final-response-level).**   You are shown: (1) the latest user query, (2) the final assistant reference response, and (3) the LLM label: `generic` / `not generic`.

- **`generic`**: the response could be pasted into many different conversations/queries with minimal edits (e.g., greetings/small talk, vague encouragement, "it depends" without specifics, generic steps not tailored to the query).

- **`not generic`**: the response provides concrete details, constraints, specific recommendations/decisions.

Mark **Correct = 1** if the LLM's `generic`/`not generic` label matches your judgment under the above definition; else **Correct = 0**.

### E.2.2. HUMAN VERIFICATION RESULTS

Table 7 shows the human verification results. GPT-4o labels match human judgments with high accuracy (about 90–96% accuracy) and strong inter-annotator consistency, supporting the reliability of our LLM-based labels for dataset-level characterization.

*Table 7.* Human verification of GPT-4o annotations (three annotators; 50 items per task and dataset). ACC is accuracy against the human majority vote. Agree is observed inter-annotator agreement. Fleiss' $\kappa$ measures agreement beyond chance.

| | MSC | | | REALTALK | | |
|---|---|---|---|---|---|---|
| **Task** | **ACC** (%) | **Agree** | **Fleiss'** $\kappa$ | **ACC** (%) | **Agree** | **Fleiss'** $\kappa$ |
| Helpful-turn label | 92.000 | 0.920 | 0.527 | 90.000 | 0.987 | 0.921 |
| Generic-response label | 95.918 | 0.973 | 0.652 | 96.000 | 0.973 | 0.653 |

## F. Analysis of Incremental ICAE Failure Modes

The incremental ICAE variant fails for structural reasons (latent drift arising from repeatedly applying a one-shot objective) whereas ICAE (append) that simply accumulates compressed context and ICAE (one-shot) that re-encodes the full available context each turn work at short and medium lengths but run out of memory on very long conversations.

ICAE is trained with a one-shot compression objective: it learns to encode a contiguous context span into a latent in a single step. In the incremental variant, however, we repeatedly apply this compressor to its own compressed outputs as the dialogue progresses. This leads to latent drift and error compounding, because the model is never trained to use or update

*already-compressed* contexts. Empirically, the response quality degrades rapidly across turns, as shown in Figure 2a.

By contrast, ICAE (append) and ICAE (one-shot) perform reasonably well at short and medium lengths, but eventually run out of memory on very long conversations as shown in Table 12 and Figure 4. In other words, ICAE (append) and ICAE (one-shot) are effective but not scalable, whereas the naive incremental application is scalable but unstable. This mismatch is exactly what motivates our design: C-DIC modifies the architecture and training scheme to support *incremental* use of compression while mitigating the catastrophic degradation observed in incremental ICAE.

## G. Multi-seed Robustness and Significance Tests

To assess robustness to random initialization and stochastic training effects, we repeat all MSC and REALTALK experiments with three random seeds (42/43/44). We report mean±std across seeds for PPL, BLEU, and ROUGE-1/2/L. For REALTALK, results are reported in the *per-session* setting due to GPU memory limits of certain compression baselines under the full long-context setup (see Figure 4 and Table 12).

### G.1. Mean±std Across Seeds

Tables 8 and 9 summarize mean±std over three runs on MSC and REALTALK, respectively. Across three independent runs, our method exhibits low run-to-run variance (e.g., PPL std $\approx 0.04$ on both datasets) while consistently outperforming the strongest baseline ICAE(one-shot) across all reported metrics.

*Table 8.* MSC results (mean±std over seeds 42/43/44).

| Models | PPL↓ | BLEU↑ | R-L↑ | R-1↑ | R-2↑ |
|---|---|---|---|---|---|
| AutoCompressor | 9.109±0.273 | 0.014±0.002 | 0.121±0.002 | 0.145±0.003 | 0.021±0.001 |
| ICAE (incremental) | 561.702±347.397 | 0.007±0.001 | 0.063±0.006 | 0.075±0.008 | 0.005±0.001 |
| ICAE (one-shot) | 29.188±1.371 | 0.017±0.000 | 0.132±0.001 | 0.188±0.002 | 0.027±0.001 |
| **Ours** | **8.385±0.042** | **0.025±0.002** | **0.159±0.001** | **0.202±0.003** | **0.037±0.000** |

*Table 9.* REALTALK results (mean±std over seeds 42/43/44). Results are in the *per-session* setting.

| Models | PPL↓ | BLEU↑ | R-L↑ | R-1↑ | R-2↑ |
|---|---|---|---|---|---|
| AutoCompressor | 12.283±0.352 | 0.020±0.001 | 0.090±0.030 | 0.138±0.003 | 0.020±0.001 |
| ICAE (incremental) | 135.827±39.254 | 0.019±0.001 | 0.071±0.003 | 0.089±0.003 | 0.013±0.001 |
| ICAE (one-shot) | 25.115±3.388 | 0.025±0.001 | 0.113±0.005 | 0.159±0.006 | 0.024±0.002 |
| **Ours** | **9.764±0.043** | **0.034±0.001** | **0.136±0.002** | **0.177±0.002** | **0.032±0.001** |

### G.2. Seed-level Significance Tests

As additional evidence that gains are not driven by a favorable seed, we compute p-values using a paired t-test on the *seed-wise differences* between our method and ICAE(one-shot) (three paired observations). Table 10 reports the resulting p-values. For PPL, we apply the test on $\log(\text{PPL})$ to reflect the likelihood (average NLL) scale. Against ICAE(one-shot), improvements are statistically significant on both datasets (all $p < 0.05$).

*Table 10.* Paired t-test p-values for **Ours** vs. ICAE(one-shot) across three seeds (42/43/44).

| Dataset | $\log(\text{PPL})$ | BLEU | R-L | R-1 | R-2 |
|---|---|---|---|---|---|
| MSC | $2.8\times10^{-4}$ | 0.010 | $9.6\times10^{-5}$ | 0.003 | $5.6\times10^{-4}$ |
| REALTALK | 0.004 | 0.002 | 0.013 | 0.027 | 0.024 |

## H. Closed-loop evaluation

This section reports additional results under *closed-loop* generation, where the assistant's past turns in the context are replaced by the model's own previously generated responses (user turns are kept fixed from the dataset). This evaluation explicitly stress-tests error accumulation under self-conditioned history.

*Table 11.* Closed-loop results on REALTALK (per-session).

| Models | PPL↓ | BLEU↑ | R-L↑ | R-1↑ | R-2↑ |
|---|---|---|---|---|---|
| Full prompting | 29.666 | 0.020 | 0.106 | 0.150 | 0.017 |
| Truncation | 28.174 | 0.021 | 0.109 | 0.156 | 0.019 |
| Summarization | 27.977 | 0.022 | 0.110 | 0.162 | 0.021 |
| In-Session RAG | 26.789 | 0.020 | 0.103 | 0.149 | 0.015 |
| AutoCompressor | 13.111 | 0.020 | 0.111 | 0.144 | 0.021 |
| ICAE (incremental) | 124.024 | 0.020 | 0.068 | 0.088 | 0.012 |
| ICAE (one-shot) | 17.364 | 0.024 | 0.109 | 0.152 | 0.023 |
| ICAE (append) | | | *Out of memory* | | |
| **Ours** | **9.754** | **0.034** | **0.133** | **0.173** | **0.031** |

Table 11 evaluates all methods on REALTALK in the *per-session* setting. We use per-session to keep methods comparable and largely runnable: several compression baselines exceed GPU memory under the full long-context (*all-sessions*) configuration, and ICAE(append) is OOM even in per-session (reported explicitly).

As shown in Table 11, our method achieves the best overall performance among runnable baselines across all reported metrics. Compared with the teacher-forcing results in Table 1, scores decrease slightly, which is expected under closed-loop generation due to compounding errors and occasional user–assistant misalignment on offline corpora. Nevertheless, the degradation for our method is modest, providing evidence that C-DIC remains stable when conditioning on its own generations.

## I. Detailed latency components

Table 12 decomposes end-to-end latency on REALTALK-*all sessions* as the maximum number of preserved turns increases ($\{10, 20, 30, 40, 428\}$). We report *Comp. Time* (time spent on context preparation such as compression/selection) and *Gen. Time* (model runtime to produce the response); *Total Time* is their sum. All values are in seconds. *Out of memory* indicates a method failed at that context length.

*Table 12.* **Latency components depending on the max number of turns.** Compression (**Comp. Time**) and generation (**Gen. Time**) times for ICAE (one-shot), ICAE (append), and C-DIC (Ours) on REALTALK at maximum turns $\{10, 20, 30, 40, 428\}$. The total latency equals **Comp. Time + Gen. Time** All values are in seconds. *Out of memory* indicates the baseline failed to run at that maximum turn due to the context length.

| Models | Max # of Turns | Comp. Time | Gen. Time | Total Time |
|---|---|---|---|---|
| Full prompting | | 0.00 | 3.53 | 3.53 |
| ICAE (one-shot) | 10 | 0.26 | 3.97 | 4.24 |
| ICAE (append) | | 0.11 | 3.53 | 3.64 |
| **Ours** | | **0.05** | **3.16** | **3.21** |
| Full prompting | | 0.00 | 4.81 | 4.81 |
| ICAE (one-shot) | 20 | | *Out of memory* | |
| ICAE (append) | | 0.27 | 3.97 | 4.24 |
| **Ours** | | **0.05** | **3.28** | **3.33** |
| Full prompting | | 0.00 | 7.66 | 7.66 |
| ICAE (one-shot) | 30 | | *Out of memory* | |
| ICAE (append) | | | *Out of memory* | |
| **Ours** | | **0.05** | **3.21** | **3.26** |
| Full prompting | | | *Out of memory* | |
| ICAE (one-shot) | 40 | | *Out of memory* | |
| ICAE (append) | | | *Out of memory* | |
| **Ours** | | **0.05** | **3.34** | **3.40** |
| Full prompting | | | *Out of memory* | |
| ICAE (one-shot) | 428 | | *Out of memory* | |
| ICAE (append) | | | *Out of memory* | |
| **Ours** | | **0.06** | **3.30** | **3.36** |

## J. Bounded Retrieval and Similarity-Scoring Overhead

C-DIC does not impose a hard bound on the total number of stored memory slots: as dialogue length grows, new or revised memory states may be added to the memory bank. However, generation depends only on the retrieved subset of memory slots, which remains small in practice. Table 13 reports memory growth and retrieval statistics across turn ranges on REALTALK and LongMemEval. On REALTALK, the average number of total slots increases from 4.07 at 1–50 turns to 62.32 at 351–400 turns, while the average number of retrieved slots remains around 2–3. Scoring time also remains small across turn ranges.

*Table 13.* **Memory growth and retrieved slot usage.** Total memory slots grow with dialogue length, while retrieved slots remain small. Scoring time is reported in milliseconds.

| # Turns | REALTALK | | | LongMemEval | | |
|---|---|---|---|---|---|---|
| | Total Slots | Retrieved Slots | Scoring (ms) | Total Slots | Retrieved Slots | Scoring (ms) |
| 1–50 | 4.07 | 1.70 | 0.5 | 3.45 | 1.47 | 0.4 |
| 51–100 | 12.08 | 2.34 | 0.3 | 6.71 | 1.99 | 0.4 |
| 101–150 | 18.88 | 2.26 | 0.3 | 10.46 | 2.11 | 0.4 |
| 151–200 | 28.20 | 2.31 | 0.3 | 14.41 | 2.15 | 0.4 |
| 201–250 | 35.91 | 2.29 | 0.3 | 17.93 | 2.19 | 0.4 |
| 251–300 | 43.52 | 2.56 | 0.3 | 19.74 | 2.09 | 0.4 |
| 301–350 | 55.24 | 1.98 | 0.3 | 22.00 | 1.92 | 0.3 |
| 351–400 | 62.32 | 2.78 | 0.4 | – | – | – |

These results show that the retrieved context used by the generator remains small. To obtain a fixed generation-side conditioning size, we additionally evaluate a bounded retrieval variant with retrieval budget $B_{\text{ret}} = 6$.

*Table 14.* **Bounded retrieval on REALTALK.** We compare the default retrieval rule with a bounded variant that caps the number of retrieved memory slots at $B_{\text{ret}} = 6$. The bounded variant gives the generator a fixed-size retrieved latent context while preserving comparable generation quality.

| Setting | PPL↓ | BLEU↑ | R-L↑ | R-1↑ | R-2↑ | Total Slots | Ret. Slots |
|---|---|---|---|---|---|---|---|
| Bounded $B_{\text{ret}} = 6$ | 9.534 | 0.026 | 0.160 | 0.226 | 0.068 | 42.900 | 2.074 |
| Default | 9.579 | 0.025 | 0.160 | 0.228 | 0.064 | 42.800 | 2.179 |

The bounded variant gives generation a fixed retrieved-context size, since the generator conditions on at most $B_{\text{ret}}$ memory slots regardless of the elapsed number of turns. However, selecting retrieved slots still requires similarity scoring over the current memory bank, whose cost scales with the total number of stored slots $N$. We therefore separately measure scoring and retrieval time as $N$ increases.

*Table 15.* **Similarity-scoring and retrieval overhead.** We measure wall-clock time for scoring $N$ memory slots and selecting the top $B_{\text{ret}}$ slots. All values are in seconds.

| $N$ | Scoring | Retrieval | Total |
|---|---|---|---|
| $10^1$ | 0.0001109 | 0.0000246 | 0.0001233 |
| $10^2$ | 0.0001117 | 0.0000274 | 0.0001313 |
| $10^3$ | 0.0001367 | 0.0000346 | 0.0001650 |
| $10^4$ | 0.0005708 | 0.0000950 | 0.0006630 |
| $10^5$ | 0.0042849 | 0.0001182 | 0.0043072 |
| $10^6$ | 0.0407054 | 0.0001791 | 0.0408903 |

In our REALTALK statistics, the memory bank contains about 62 slots by turn 400; under a conservative linear extrapolation, $N = 10^3$ would correspond to roughly 6.4k dialogue turns. At this scale, scoring and top-$B_{\text{ret}}$ retrieval take only 0.16ms, compared with roughly 3s generation time, and even $N = 10^6$ incurs only 41ms overhead.

## K. Sensitivity to Retrieval Threshold and Decay

C-DIC uses a retrieval threshold $\tau$ to decide whether a memory state should be reused and revised, and a decay rate $\alpha$ to control recency bias in retrieval. Although $\tau$ is not directly optimized, it interacts with learned representations: the encoder

and memory updater shape query–memory cosine similarities during training, making a fixed threshold a stable retrieval boundary. Since $\tau$ and $\alpha$ affect both response quality and memory usage, we evaluate their sensitivity with generation metrics, total slot counts, and retrieved slot counts.

**Retrieval threshold $\tau$.** Table 16 shows that C-DIC is stable over a moderate range of thresholds. On MSC, performance remains similar for $\tau \in \{0.6, 0.7, 0.8\}$, while a stricter threshold $\tau = 0.9$ substantially increases the number of total slots and degrades quality. This is consistent with the expected behavior: overly strict retrieval prevents the model from revising existing memory states and instead encourages adding new ones. LongMemEval shows a similar trend: $\tau = 0.8$ provides the best accuracy among the tested thresholds, while $\tau = 0.9$ sharply increases memory growth.

*Table 16.* **Sensitivity to retrieval threshold $\tau$.** We report generation quality and memory usage on MSC, evaluated on MSC-session 5 at the final dialogue turn, and answer accuracy and memory usage on LongMemEval.

| | MSC | | | | | | | LongMemEval | | |
|---|---|---|---|---|---|---|---|---|---|---|
| $\tau$ | PPL$\downarrow$ | BLEU$\uparrow$ | R-L$\uparrow$ | R-1$\uparrow$ | R-2$\uparrow$ | Total Slots | Ret. Slots | Acc.$\uparrow$ | Total Slots | Ret. Slots |
| 0.6 | 8.325 | 0.028 | 0.157 | 0.208 | 0.037 | 3.022 | 1.000 | 0.104 | 1.004 | 1.002 |
| 0.7 | 8.383 | 0.027 | 0.156 | 0.207 | 0.036 | 3.022 | 1.000 | 0.114 | 1.362 | 1.064 |
| 0.8 | 8.427 | 0.030 | 0.160 | 0.206 | 0.040 | 3.501 | 1.015 | 0.116 | 18.762 | 1.980 |
| 0.9 | 12.202 | 0.022 | 0.139 | 0.190 | 0.027 | 30.026 | 1.050 | 0.114 | 160.658 | 1.242 |

**Decay rate $\alpha$.** Table 17 shows that C-DIC is also robust to moderate changes in the decay rate. On MSC, the three tested values yield similar generation quality and slot usage. On LongMemEval, increasing $\alpha$ reduces both total and retrieved slot counts, and $\alpha = 0.1$ achieves the highest accuracy among the tested values. This suggests that a moderate recency bias can reduce retrieval breadth without harming long-range performance.

*Table 17.* **Sensitivity to decay rate $\alpha$.** We report generation quality and memory usage on MSC, evaluated on MSC-session 5 at the final dialogue turn, and answer accuracy and memory usage on LongMemEval.

| | MSC | | | | | | | LongMemEval | | |
|---|---|---|---|---|---|---|---|---|---|---|
| $\alpha$ | PPL$\downarrow$ | BLEU$\uparrow$ | R-L$\uparrow$ | R-1$\uparrow$ | R-2$\uparrow$ | Total Slots | Ret. Slots | Acc.$\uparrow$ | Total Slots | Ret. Slots |
| 0 | 8.399 | 0.025 | 0.160 | 0.207 | 0.036 | 3.150 | 1.005 | 0.098 | 23.920 | 2.810 |
| 0.05 | 8.427 | 0.030 | 0.160 | 0.206 | 0.040 | 3.501 | 1.015 | 0.116 | 18.762 | 1.980 |
| 0.1 | 8.353 | 0.026 | 0.157 | 0.207 | 0.038 | 3.022 | 1.000 | 0.120 | 5.014 | 1.478 |

Overall, these sweeps show that C-DIC is not highly sensitive to moderate choices of $\tau$ and $\alpha$. The main failure mode appears when retrieval is overly strict: the model stops revising existing memory states, leading to unnecessary memory growth and degraded quality. Moderate recency bias can further reduce retrieval breadth, especially on LongMemEval, while maintaining or improving accuracy.

## L. Effect of compression token length

This work studies *incremental* compression for multi-turn dialogue by extending ICAE (Ge et al., 2024), rather than re-optimizing the underlying compressor for static-document compression. Accordingly, unless otherwise stated, our main experiments use the publicly released ICAE checkpoint, which provides a single compression length (128 tokens).

To examine the impact of compression length in the multi-turn setting, we train ICAE-style compressors with 64, 128, and 256 tokens on MSC (Xu et al., 2022) using the same ICAE training objectives (one-shot continuation and auto-encoding). We then fine-tune the dialogue model with our proposed method while varying only the compression token budget. Table 18 reports results on MSC under the 5-session evaluation setting.

These results indicate that performance is relatively stable across 64–256 tokens, with differences in PPL and generation metrics being modest and without a clear monotonic trend. This suggests that C-DIC is not overly sensitive to the exact compression capacity within this range.

*Table 18.* **Comparison of compression token lengths on MSC-session 5.** Performance as a function of the selection threshold $\tau$. We report PPL↓, BLEU↑, ROUGE-L↑, ROUGE-1↑, and ROUGE-2↑.

| Comp. Token Length | PPL↓ | BLEU↑ | R-L↑ | R-1↑ | R-2↑ |
|---|---|---|---|---|---|
| 64 | 8.604 | 0.023 | 0.157 | 0.201 | 0.036 |
| 128 | 8.582 | 0.023 | 0.155 | 0.200 | 0.034 |
| 256 | 8.646 | 0.022 | 0.160 | 0.205 | 0.037 |

*Table 19.* **LongMemEval results (zero-shot).** All methods use the same backbone (LLaMA2-7B). Accuracy is computed by a GPT-4o judge following the LONGMEMEVAL evaluation protocol.

| Models | Accuracy↑ |
|---|---|
| Full prompting | 0.086 |
| ICAE (incremental) | 0.010 |
| ICAE (one-shot) | 0.004 |
| **Ours** | **0.116** |

## M. LongMemEval: Long-Context QA Evaluation

This appendix reports an additional evaluation on LONGMEMEVAL$_S$ (Wu et al., 2025), a benchmark designed to assess long-term memory and question answering in chat assistants. We evaluate in a **zero-shot** setting using the same LLM backbone across all methods (Llama-2-Chat-7B). We compare (i) full prompting and (ii) latent-compression baselines (ICAE variants), against (iii) our C-DIC-based approach. Following the LONGMEMEVAL protocol, we use **GPT-4o** as the automatic judge to determine answer correctness and report Accuracy.

Table 19 shows that C-DIC yields the best accuracy among the compared methods, improving over full prompting while using substantially fewer input tokens. These results provide evidence that C-DIC improves QA performance under long-context settings, complementing our dialogue-generation results on MSC and REALTALK.

## N. REALTALK: Two-Session Evaluation

The REALTALK (Lee et al., 2025) dataset contains substantially longer multi-session dialogues than MSC (Xu et al., 2022). In Table 1, we therefore report results *per session* to avoid out of memory issue. For completeness, Table 20 reports performance on the subset of REALTALK conversations with **two sessions**. We follow the same evaluation protocol and hyperparameters as in the main results. Note that our model shows consistent performance without the session limit in Figure 5.

*Table 20.* **REALTALK two-session results.** Test performance on conversations with up to two sessions. Lower is better for PPL; higher is better for BLEU/ROUGE.

| Models | PPL↓ | BLEU↑ | R-L↑ | R-1↑ | R-2↑ |
|---|---|---|---|---|---|
| Full prompting | 27.225 | 0.022 | 0.109 | 0.159 | 0.020 |
| Truncation | 21.865 | 0.026 | 0.109 | 0.174 | 0.028 |
| Summarization | 26.120 | 0.023 | 0.118 | 0.172 | 0.025 |
| In-Session RAG | 27.435 | 0.019 | 0.100 | 0.145 | 0.014 |
| AutoCompressor | 11.150 | 0.020 | 0.112 | 0.146 | 0.022 |
| ICAE (incremental) | 218.103 | 0.017 | 0.059 | 0.073 | 0.007 |
| ICAE (one-shot) | | | *Out of memory* | | |
| ICAE (append) | | | *Out of memory* | | |
| **Ours** | **9.870** | **0.034** | **0.132** | **0.175** | **0.029** |

## O. Additional Results on Multi-Session Chat (MSC)

Table 21 reports a detailed breakdown of model quality setting across different session lengths (2–5) as well as the aggregate over all sessions. We evaluate generation with perplexity (PPL; lower is better) and text-overlap metrics (BLEU, ROUGE-L/1/2; higher is better).

*Table 21.* Comparison across MSC sessions. Lower is better for PPL; higher is better for others.

| Models | Session | PPL↓ | BLEU↑ | R-L↑ | R-1↑ | R-2↑ |
|---|---|---|---|---|---|---|
| Full prompting | AVG | 41.245 | 0.008 | 0.110 | 0.157 | 0.015 |
| Truncation | | 30.890 | 0.012 | 0.128 | 0.184 | 0.024 |
| Summarization | | 41.849 | 0.013 | 0.128 | 0.172 | 0.024 |
| In-Session RAG | | 35.530 | 0.008 | 0.110 | 0.148 | 0.014 |
| AutoCompressor | | 9.285 | 0.012 | 0.121 | 0.174 | 0.021 |
| ICAE (incremental) | | 513.774 | 0.006 | 0.057 | 0.069 | 0.005 |
| ICAE (one-shot) | | 27.656 | 0.017 | 0.133 | 0.190 | 0.027 |
| **Ours** | | **8.431** | **0.023** | **0.160** | **0.205** | **0.037** |
| Full prompting | 5 | 40.801 | 0.012 | 0.113 | 0.165 | 0.016 |
| Truncation | | 26.252 | 0.014 | 0.136 | 0.198 | 0.028 |
| Summarization | | 38.9759 | 0.0148 | 0.129 | 0.1881 | 0.024 |
| In-Session RAG | | 33.931 | 0.009 | 0.112 | 0.163 | 0.014 |
| AutoCompressor | | 9.364 | 0.012 | 0.123 | 0.151 | 0.020 |
| ICAE (incremental) | | 442.062 | 0.006 | 0.061 | 0.074 | 0.004 |
| ICAE (one-shot) | | 30.312 | 0.016 | 0.134 | 0.196 | 0.027 |
| **Ours** | | **8.553** | **0.024** | **0.160** | **0.211** | **0.038** |
| Full prompting | 4 | 40.621 | 0.009 | 0.109 | 0.157 | 0.014 |
| Truncation | | 26.427 | 0.013 | 0.133 | 0.188 | 0.024 |
| Summarization | | 40.162 | 0.012 | 0.130 | 0.190 | 0.024 |
| In-Session RAG | | 33.318 | 0.008 | 0.110 | 0.161 | 0.014 |
| AutoCompressor | | 9.222 | 0.012 | 0.129 | 0.150 | 0.020 |
| ICAE (incremental) | | 454.005 | 0.006 | 0.061 | 0.074 | 0.004 |
| ICAE (one-shot) | | 29.261 | 0.016 | 0.133 | 0.194 | 0.028 |
| **Ours** | | **8.418** | **0.022** | **0.158** | **0.205** | **0.037** |
| Full prompting | 3 | 40.221 | 0.009 | 0.110 | 0.157 | 0.014 |
| Trunc.-5 | | 27.678 | 0.014 | 0.127 | 0.192 | 0.020 |
| Summarization | | 42.804 | 0.012 | 0.127 | 0.184 | 0.024 |
| In-Session RAG | | 35.751 | 0.008 | 0.110 | 0.158 | 0.018 |
| AutoCompressor | | 9.222 | 0.012 | 0.123 | 0.150 | 0.020 |
| ICAE (incremental) | | 505.368 | 0.006 | 0.058 | 0.070 | 0.004 |
| ICAE (one-shot) | | 26.971 | 0.018 | 0.137 | 0.195 | 0.029 |
| **Ours** | | **8.350** | **0.023** | **0.162** | **0.206** | **0.037** |
| Full prompting | 2 | 43.323 | 0.003 | 0.109 | 0.153 | 0.013 |
| Truncation | | 29.414 | 0.013 | 0.133 | 0.188 | 0.023 |
| Summarization | | 45.456 | 0.012 | 0.126 | 0.183 | 0.024 |
| In-Session RAG | | 39.120 | 0.008 | 0.108 | 0.148 | 0.017 |
| AutoCompressor | | 9.333 | 0.011 | 0.110 | 0.149 | 0.017 |
| ICAE (incremental) | | 653.668 | 0.006 | 0.058 | 0.070 | 0.004 |
| ICAE (one-shot) | | 24.081 | 0.018 | 0.130 | 0.174 | 0.025 |
| **Ours** | | **8.404** | **0.022** | **0.159** | **0.198** | **0.037** |

# P. Qualitative Examples

We present the qualitative examples demonstrating C-DIC's effectiveness in context coherence in a long dialogue setting in Figure 6 and 7.

*Figure 6.* **Qualitative example #1 (LongMemEval).** Multi-session dialogue in chronological order (previous sessions → current session).

---

**Dialogue Snippet**

---

**Session 1∼4**

. . . *(14 turns)* . . .

**Session 5**

. . . *(1 turn)* . . .

**Turn #16**

**S1:** I love these ideas! I'm definitely going to consider the Global Street Food theme. By the way, I've also had a great experience with a BBQ theme, like ==the one we had at Mike's place two weeks ago==, where we watched a football game together. Do you have any BBQ recipe ideas that would fit well with the Global Street Food theme?

**S2:** A BBQ theme can be a fantastic way to incorporate grilled and smoky flavors into your Global Street Food potluck. Here are some BBQ-inspired recipes . . . *(continued)*

**S1:** That's great! I think I'll have a mix of grilled and non-grilled dishes to cater to different tastes. By the way, I've also had experience with dinner parties that are more low-key, like ==the ones we had at Alex's place yesterday==, where we had a potluck and tried out different cuisines from around the world, and ==also at Mike's place==, where we had a BBQ and watched a football game together. Do you have any tips on how to make the party more interactive and engaging for guests?

**S2:** You're referring to creating a more immersive and engaging experience for your guests. . . . *(continued)*

. . . *(2 turns)* . . .

**Session 6∼8**

. . . *(8 turns)* . . .

**Session 9**

**Turn #28**

**S1:** I'm looking for some Italian recipe ideas for a dinner party I'm hosting soon. ==I attended a lovely Italian feast at Sarah's place last week==, and it inspired me to try out some new dishes. Do you have any recommendations for a beginner like me?

**S2:** Buon appetito! I'm thrilled to hear that Sarah's Italian feast inspired you to host your own dinner party. Italian cuisine is incredibly diverse and rich, so I'd be happy to help you with some recipe ideas perfect for a beginner. Here are some classic and delicious Italian dishes . . . *(continued)*

**S1:** That's a great list of recipes! I think I'll start with the Spaghetti Aglio e Olio and Chicken Parmesan. I've always loved the combination of garlic and olive oil. By the way, speaking of dinner parties, ==I've had a lovely experience at Sarah's place recently==, where we played board games until late into the night after the Italian feast. Do you have any recommendations for board games that would be a good fit for a dinner party?

**S2:** Spaghetti Aglio e Olio and Chicken Parmesan are both delicious and crowd-pleasing options. Now, about board games for a dinner party . . . *(continued)*

. . . *(4 turns)* . . .

**Session 10∼44**

. . . *(190 turns)* . . .

---

**Question:** How many dinner parties have I attended in the past month?

---

**Full prompting:** "I'm just an AI, I don't have access to your personal information or calendar, so I'm unable to accurately answer your question. However, I can suggest that you check your calendar or contact the hosts of any dinner parties you may have attended in the past month . . ."

**ICAE (incremental):** "The number of the month? Is that 30! And if you're a math teacher you'll have 30 students!"

**ICAE (one-shot):** " Reduce, Reuse, Recycle: Implement the 3Rs in your daily life by reducing your consumption, . . ."

**Ours:** "You've attended **3 dinner parties** in the past month."

**Ground-truth:** "Three"

---

*Figure 7.* **Qualitative example #2 (LongMemEval).** Multi-session dialogue in chronological order (previous sessions → current session).

| Dialogue Snippet |
| --- |

**Session 1∼11**

. . . *(56 turns)* . . .

**Session 12**

. . . *(1 turn)* . . .

**Turn #58**

**S1:** I'm also thinking of exploring other platforms like Instagram and Twitter to promote my writing services. Do you have any tips on how to get started with those platforms, especially since I don't have much experience with them? By the way, speaking of first purchases, I remember getting a **10% discount** on my first purchase from that new clothing brand last month, which was nice.

**S2:** Great mention about the discount! Who doesn't love a good deal, right? Now, let's dive into getting started with Instagram and Twitter to promote your writing services. Both platforms have unique features and audiences, so I'll provide you with some tips to help you get started: **Instagram:** 1. **Visual content is key**: Instagram is all about visuals, so focus on creating high-quality images, graphics, or videos that showcase your writing skills, books, or writing process. 2. **Use hashtags strategically**: Research and use relevant hashtags in your captions to increase visibility. You can use a mix of niche-specific hashtags (e.g., #writingcommunity, #freelancewriting) and broader hashtags (e.g., #writerslife, #booklovers). 3. **Create an Instagram Stories strategy**: Share behind-the-scenes content, sneak peeks, or Q&A sessions to engage with your audience. You can also use the "swipe-up" feature to drive traffic to your website or other social media profiles. 4. **Leverage Instagram's features**: Use IGTV for longer-form content, Reels for short videos, and IG Live for real-time engagement. 5. **Collaborate with other writers or influencers**: Partner with fellow writers or influencers in your niche to reach new audiences and create content that appeals to a broader range of people. 6. **Post consistently**: Aim for at least 3-5 posts per week, but make sure you don't sacrifice quality for quantity. 7. **Engage with your audience**: Respond to comments, answer questions, and use Instagram's features like polls and quizzes to encourage interaction. **Twitter:** 1. **Keep it concise**: Twitter is all about short, bite-sized content. Keep your tweets concise, informative, and engaging. 2. **Use hashtags**: Research and use relevant hashtags in your tweets to increase visibility. You can use a mix of niche-specific hashtags (e.g., #writingtips, #freelancewriting) and broader hashtags (e.g., #writerslife, #books). 3. **Tweet regularly**: Aim for at least 5-10 tweets per week, but make sure you don't overwhelm your audience. 4. **Engage with others**: Respond to tweets, answer questions, and participate in Twitter Chats to build relationships with other writers, influencers, and potential clients. 5. **Share a mix of content**: Share your own content, retweet relevant articles or tweets, and participate in Twitter polls to keep your audience engaged. 6. **Use Twitter Ads**: Consider running targeted ads to increase your reach and engagement, especially if you're just starting out. 7. **Optimize your profile**: Make sure your Twitter profile is complete, including a profile picture, header image, and bio that clearly states your writing services and expertise. **Additional Tips:** 1. **Cross-promote**: Share your Instagram and Twitter content on your other social media platforms, like Facebook and LinkedIn, to reach a broader audience. 2. **Be authentic**: Be yourself, and let your personality shine through on both platforms. This will help you build a loyal following and attract potential clients who resonate with your voice and style. 3. **Monitor and adjust**: Keep an eye on your analytics and adjust your strategy based on what's working and what's not. Remember, building a presence on Instagram and Twitter takes time and effort. Focus on creating high-quality content, engaging with your audience, and promoting your writing services consistently. Good luck!

. . . *(4 turn)* . . .

**Session 13∼48**

. . . *(187 turns)* . . .

| |
| --- |
| **Question:** What was the discount I got on my first purchase from the new clothing brand? |
| **Full prompting:** "I'm just an AI, I don't have access to personal information or purchase history, . . ." |
| **ICAE (incremental):** "I am really excited to see what kind of discount you got given on your first purchase from the new clothing brand!" |
| **ICAE (one-shot):** "southeast asian cultures: southeast asian cultures have a rich history of textile and print design, . . ." |
| **Ours:** "You got **10 %** off your first purchase!" |
| **Ground-truth:** "10 %" |

