# OpenReview forum: "Context-Driven Incremental Compression for Multi-Turn Dialogue Generation"
_ICML.cc/2026/Conference — ICML 2026 regular_

### Official Review · Reviewer_KfLc · 2026-03-12

**Soundness:** 2
**Presentation:** 2
**Significance:** 3
**Originality:** 2
**Overall Recommendation:** 4
**Confidence:** 4

**Summary:**

This paper proposes Context-Driven Incremental Compression (C-DIC) for long multi-turn dialogue generation. The method maintains a compact memory of thread-level compressed states and updates it turn by turn via a retrieve → compress/revise → write-back loop. The compressor is initialized from ICAE, the generator is frozen, and training uses a retrieval-aware truncated BPTT scheme. Experiments on MSC and REALTALK show improvements over truncation, summarization, RAG, AutoCompressor, and ICAE variants, with additional latency and ablation results.

**Compliance With Llm Reviewing Policy:**

Affirmed.

**Final Justification:**

My final recommendation is **positive / weak accept**.

This paper studies an important problem—scalable long-horizon multi-turn dialogue modeling—and proposes a coherent approach that combines incremental latent compression, retrieval-conditioned memory updates, and retrieval-aware truncated credit assignment. The work’s main strengths are the relevance of the problem, the practical design of the method, and a substantial empirical package including main benchmark results, latency analysis, closed-loop evaluation, ablations, and additional long-context QA evidence. Even in the original submission, the paper had clear merit; my hesitation was primarily about whether some of the stronger claims were fully supported and whether key parts of the technical exposition were stated precisely enough.

The rebuttal addressed my main concerns and changed my evaluation. In particular, the authors clarified the training/inference distinction ($r_t$ during teacher-forced training versus $\hat r_t$ at inference), clarified that Eq. 8 is an intentional sparse-credit approximation with stop-gradient on non-argmax retrieved supports, and committed to making this explicit in the revision. These points resolved my methodological concerns about internal consistency and the status of the gradient rule.

The rebuttal also substantially strengthened the efficiency and long-range-fidelity evidence. On efficiency, the added bounded (k=6) variant is useful because it shows that the model can preserve essentially the same quality while yielding a clear bounded-conditioning interpretation for per-turn generation cost. The added scaling measurements for similarity scoring and top-(k) retrieval also make the practical overhead story much more convincing. On long-range fidelity, expanding MSC-QA to the full set provides a much more direct and systematic evaluation of reference tracking on the main dialogue benchmark, and the results are consistent with the broader evidence already present in Appendix E, Appendix L, Appendix O, and the closed-loop experiments. Taken together, this makes the paper’s core empirical story significantly stronger than in the original version.

I still see some limitations. The originality is solid but moderate: the paper is not introducing an entirely new paradigm so much as a well-motivated integration of compression, retrieval, memory update, and truncated credit assignment for the dialogue setting. The current draft also overstates some claims and needs clearer positioning relative to prior memory/compression work. In addition, the presentation should be tightened, especially around notation and the exact training/inference behavior. However, after the rebuttal, these issues look more like revision-quality concerns than reasons to reject the paper.

Overall, I now believe the paper makes a meaningful contribution. The problem is important, the method is technically reasonable, the empirical evidence is now much stronger, and the rebuttal directly addressed the issues that initially drove my negative score. Assuming the revision incorporates the promised clarifications, moderates wording around novelty/efficiency, and integrates the new bounded-efficiency and full-scale MSC-QA results into the paper, I support acceptance.

**Key Questions For Authors:**

1. How does memory size grow with dialogue length?
   Please report the number of active memory slots as turns increase and ideally include a fixed-budget memory variant. It directly determines whether the efficiency claims are actually supported.
2. What exactly is compressed during training and inference: rt or rt ? The paper and Algorithm 1 are not fully consistent here.
3. As written, Eq. (8) does not appear to be the exact gradient of the forward graph as stated, because the current-turn loss depends on all retrieved slots in ℛt, and the new state Zt is also computed from all retrieved supports. Please clarify whether non-argmax retrieved slots are explicitly detached in implementation, or whether Eq. (8) should be interpreted as an intentional truncated-credit approximation rather than an exact consequence of the chain rule.
4. The appendix already includes a LongMemEval evaluation (Appendix L) and qualitative long-range fact-recall examples (Appendix O), which I appreciate. However, the main claims of the paper concern long-horizon dialogue generation and thread-level contextual fidelity, while the main benchmark evidence on MSC/REALTALK still relies primarily on PPL/BLEU/ROUGE. Could the authors provide a more direct and systematic evaluation of long-range reference fidelity on the main dialogue benchmarks (e.g., targeted reference-tracking QA, entity/slot recall, or human/LLM judgments of factual consistency across turns), and clarify how the LongMemEval results should be interpreted relative to the core dialogue-generation claims?

**Limitations:**

No. The paper should more explicitly discuss memory growth, dependence on a specific pretrained compressor / frozen-backbone setting, limited evaluation breadth, and privacy / stale-memory risks in persistent dialogue memory.

**Strengths And Weaknesses:**

This paper addresses an important problem, and the empirical observation that static compressors degrade under multi-turn rollout is interesting. The latency analysis and closed-loop experiments are also useful additions.

However, I have substantial concerns.

On soundness, several central claims are stronger than what the paper actually supports. The method is presented as having near-constant inference cost, but memory size is not bounded in the algorithm: new slots are inserted under topic shift, and retrieval appears to score against all existing slots. This makes the “near-constant” narrative incomplete unless memory growth is shown to remain controlled in practice. In addition, the training rule is not fully aligned with the forward computation: multiple slots can be retrieved in ℛt, but gradients appear to flow only through the argmax slot. This is a strong approximation, and the paper does not justify it well. There are also methodological inconsistencies, especially around whether compression uses rt or rt, and the equations / algorithm are not fully synchronized.

On presentation, the paper is weaker than it should be for a top-tier venue. There are many writing issues, overstatements, and notation inconsistencies. Several claims are phrased too aggressively (“first framework”, “state-of-the-art”, “near-constant inference time”) given the actual evidence. The method can be followed at a high level, but the technical exposition is not clean enough.

On significance, the topic is clearly important, and the idea may be practically useful. That said, the experimental evidence is not yet broad enough to establish strong impact. The paper relies heavily on PPL/BLEU/ROUGE, mostly uses a single frozen Llama-2-Chat-7B setup, and some comparisons are weakened by OOM constraints or altered evaluation settings. I do not think the current experiments fully support the broader claims.

On originality, the work is moderately novel but not highly original. The main contribution is an integration of existing ingredients—latent compression, retrieval, memory update, and truncated credit assignment—adapted to multi-turn dialogue. That can still be valuable, but the paper does not position itself sharply enough relative to prior memory/compression methods, so the novelty feels narrower than claimed.

Overall, this is a promising paper with a good problem setting and some encouraging results, but the current version is not technically tight enough for acceptance.

---

> ### Author Rebuttal · Authors · 2026-03-31
>
> ### 1) Unbounded memory growth weakens the 'near-constant inference time' claim
> >Thank you for highlighting this important point. We will clarify that “near-constant inference time” is an **empirical observation**, not a formal O(1) guarantee. We therefore add **memory-growth and retrieval-cost statistics** on long dialogues. On **REALTALK**, as dialogue length increases from 1–50 to 351–400 turns, average total slots grows from **4.07 to 62.32, but retrieved slots remain small, and scoring time stays nearly flat**. **LongMemEval** shows the same trend. Thus memory is **not formally bounded, but latency remains stable in practice**.
> >
> >**REALTALK**
> >|# of Turns|Total Slots|Retrieved Slots|Scoring Time|
> >|---|---|---|---|
> >|1–50|4.0740|1.7020|0.0005|
> >|51–100|12.0777|2.3440|0.0003|
> >|101–150|18.8790|2.2645|0.0003|
> >|151–200|28.2006|2.3083|0.0003|
> >|201–250|35.9067|2.2867|0.0003|
> >|251–300|43.5233|2.5614|0.0003|
> >|301–350|55.2400|1.9800|0.0003|
> >|351–400|62.3233|2.7800|0.0004|
> >
> > **LongMemEval**
> >|# of Turns|Total Slots|Retrieved Slots|Scoring Time|
> >|---|---|---|---|
> >|1–50|3.4480|1.4721|0.0004|
> >|51–100|6.7060|1.9914|0.0004|
> >|101–150|10.4600|2.1126|0.0004|
> >|151–200|14.4060|2.1537|0.0004|
> >|201–250|17.9339|2.1936|0.0004|
> >|251–300|19.7443|2.0935|0.0004|
> >|301–350|22.0000|1.9167|0.0003|
> ### 2) Methodological inconsistencies: what is compressed during training vs inference: $r_t$ or $\hat r_t$?
> >This difference is due to the **training vs. inference setting**. Training is **teacher-forced**: when forming $Z_t$, the compressor uses **gold response $r_t$**. At inference, the same step uses **model-generated response $\hat r_t$**, since the reference is unavailable. The main text describes training while Algorithm 1 describes inference. We will make this explicit next to Eq. 5 and in Algorithm 1.
> ### 3) Eq. 8 is not the exact gradient if multiple slots are retrieved
> >Thank you for your valuable comment. Eq. 8 is not the exact gradient of the unrestricted multi-slot graph. Instead, it is an intentional **sparse-credit approximation**: although forward pass may retrieve multiple slots, write-back step updates only the **argmax-linked slot $Z_{j_t}$**. We thus backpropagate only through that updated slot and stop gradients through the other retrieved supports. This matches training to the **actual memory-update path**. We will revise the paper to make this motivation explicit.
> ### 4) Experimental evidence is not broad enough yet
> >
> >**4.1) Long-range fidelity**
> >
> >Thank you for this constructive feedback. We have added a targeted reference-tracking QA evaluation (see our response to **Reviewer Svrw, Q2**).
> >
> >**4.2) Mostly one frozen Llama-2-Chat-7B setup**
> >
> >To reduce dependence on a single backbone, we added Llama-3.1-8B results against Full Prompting (see our response to **Reviewer k3Xd, Q2**).
> >
> >**4.3)  Some comparisons are weakened by OOM constraints or altered evaluation settings**
> >
> >This is **not due to altered evaluation settings**: all methods are evaluated under the **same backbone and hardware budget**. In very long dialogues, some baselines become infeasible because their memory usage grows with dialogue length, leading to **OOM even with batch size 1 on an A100 80GB GPU**. Since a core contribution of our paper is **scalable long-term dialogue modeling**, feasibility under a shared budget is an important part of the comparison.
> ### 5) Novelty claims and wording should be moderated
> >We clarify that the main contribution of C-DIC is a **dialogue-specific operational view of compression**: instead of compressing history once as static context, C-DIC maintains a **persistent latent memory** that is **incrementally retrieved, revised, and written back** across turns. Prior compression methods are mainly **static/one-shot**, while prior memory/retrieval approaches do not provide the same **revisable latent dialogue-memory formulation with retrieval-aware training**. In C-DIC, multi-slot retrieval, incremental write-back, and retrieval-aware truncated credit assignment are integrated to match the actual memory-update path in dialogue. We will revise the paper to sharpen this positioning and moderate all mentioned wordings.
> ### 6) Discussion of limitations: memory growth, pretrained-compressor dependence, stale memory, limited evaluation breadth, and privacy/stale-memory risks
> >Thank you for this suggestion. We will add an explicit limitations discussion. Memory growth: although C-DIC keeps retrieval cost small in our experiments, memory can still grow when the dialogue repeatedly shifts topics. Dependence on pretrained components / frozen-backbone setting: the current formulation relies on a pretrained ICAE-style compressor and a frozen generator. Limited evaluation breadth: despite adding stronger baselines and MSC-QA, our experiments still cover a limited set of benchmarks and diagnostics. Privacy / stale-memory risks: persistent latent memory may retain sensitive information and propagate stale.

---

> > ### Author Rebuttal · Reviewer_KfLc · 2026-04-02
> >
> > Thank you for the detailed rebuttal. The added memory-growth and retrieval-cost statistics are helpful and clarify that the “near-constant inference time” claim should be understood as an empirical observation rather than a formal bounded-cost guarantee. The clarification that training uses teacher forcing with $r^t$ while inference uses $\hat{r}^t$ also resolves the inconsistency I noted between the equations and Algorithm 1. In addition, the explanation that Eq. 8 is an intentional sparse-credit approximation, with gradients stopped through non-argmax retrieved supports, is useful and improves my understanding of the method.
> >
> > The added MSC-QA reference-tracking results and the comparison against stronger full prompting with a longer-context backbone also strengthen the empirical case. My remaining concern is that the efficiency/scalability story is still empirical rather than formally bounded, and the long-range fidelity evidence on the main dialogue task, while improved, is still somewhat limited in scope. In the revision, please make the sparse-credit approximation and stop-gradient behavior explicit in the main text and algorithm, integrate the new scaling and fidelity results directly into the paper, and moderate claims such as “near-constant inference time” and the stronger novelty wording accordingly.

---

> > > ### Author Response · Authors · 2026-04-05
> > >
> > > Thank you for the thoughtful engagement. We are glad that our clarifications on the training/inference distinction and sparse-credit approximation have fully resolved your methodological concerns. We also appreciate your recognition that the memory-growth and retrieval-cost statistics support C-DIC's efficiency, and that the MSC-QA and stronger-backbone results further strengthen our key claims. As reflected in the response, the remaining concerns are not about the core approach, but about the formal efficiency bound and long-range fidelity evidence. We address both below.
> > >
> > > ### 1) Formally bounded efficiency
> > > We appreciate your acknowledgement of our stable latency in practice. To further address your concern, we first analyze slot usage on REALTALK and find that >95% of turns use ≤6 slots. Motivated by this, we **additionally** evaluate a bounded variant of C-DIC with a hard cap of $k=6$ retrieved slots.
> > > ||PPL ↓|BLEU ↑|R-L ↑|R-1 ↑|R-2 ↑|Total Slots|Retrieved Slots
> > > ---|---|---|---|---|---|---|---
> > > Bounded $k=6$|9.534|0.026|0.160|0.226|0.068|42.900|2.074
> > > Default|9.579|0.025|0.160|0.228|0.064|42.800|2.179
> > >
> > > The bounded variant shows comparable performance, while yielding formal O(1) per-turn cost w.r.t. dialogue length, since the generator takes at most $k$ memory slots regardless of elapsed turns.
> > >
> > > A separate concern is similarity scoring cost, which scales with total number of memory slots $N$. To quantify this, we **additionally** measure scoring & top-$k$ retrieval times as $N$ grows:
> > > N|scoring_s|retrieval_s|total_s
> > > ---|---|---|---
> > > $ 10^1 $|0.0001109|0.0000246|0.0001233
> > > $ 10^2 $|0.0001117|0.0000274|0.0001313
> > > $ 10^3 $|0.0001367|0.0000346|0.0001650
> > > $ 10^4 $|0.0005708|0.0000950|0.0006630
> > > $ 10^5 $|0.0042849|0.0001182|0.0043072
> > > $ 10^6 $|0.0407054|0.0001791|0.0408903
> > >
> > > Both scoring and top-$k$ retrieval incur negligible overhead (0.16ms at N=1k vs. ~3s generation). Our retrieval-cost statistics show ~62 slots by turn 400; N=1k implies ~6.4k dialogue turns under conservative linear extrapolation, far beyond realistic conversations. Even at N=1M, the overhead is only 41ms.
> > >
> > > In summary, C-DIC's efficiency can be guaranteed with a simple retrieval cap.
> > >
> > > ### 2) Long-range fidelity on the main dialogue task
> > > We appreciate the reviewer’s constructive suggestions and agree on the importance of long-range fidelity. Extensive experiments addressing this concern were **already** conducted in the original paper (summarized below). To further strengthen the evidence, we **expand** MSC-QA from 50 samples to the full set, which consistently supports our conclusions.
> > >
> > > Specifically, Appendix E confirms that 70.8% of MSC and 40.3% of REALTALK dialogues require context ≥10 turns back, with only 2–6% generic targets — strong performance cannot be explained by short-range cues alone. Appendix O shows only C-DIC correctly aggregating facts across hundreds of intervening turns. LongMemEval (Appendix L) independently confirms this advantage on a separate long-context QA benchmark.
> > >
> > > To further address your concern, we scale MSC-QA to the full set, directly measuring reference tracking on the main benchmark. All models are trained solely for generation with no QA supervision, so absolute accuracy reflects zero-shot transfer:
> > > ||Accuracy|PPL ↓|BLEU ↑|R-L ↑|R-1 ↑|R-2 ↑
> > > ---|---|---|---|---|---|---
> > > Full Prompting|0.042|30.560|0.016|0.121|0.160|0.027
> > > ICAE (incremental)|0.000|821.209|0.009|0.057|0.066|0.004
> > > ICAE (oneshot)|0.004|38.268|0.010|0.095|0.125|0.012
> > > InfLLM|0.062|8.963|0.020|0.146|0.181|0.040
> > > Ours|**0.104**|**5.950**|**0.039**|**0.205**|**0.243**|**0.064**
> > >
> > > C-DIC achieves the highest accuracy and best generation quality across all metrics, consistent with our subset results. Higher reference-tracking accuracy co-occurs with better generation metrics, suggesting that generation quality is tied to long-range context utilization, even outperforming Full Prompting with access to the entire raw history.
> > >
> > > Together — the dataset characterization (Appendix E), qualitative multi-hop retrieval (Appendix O), LongMemEval (Appendix L), closed-loop stability over 400+ turns, and now full-scale MSC-QA — these provide strong converging evidence that C-DIC's improvements are driven by superior long-range context utilization.
> > >
> > > ### Revision plan
> > > Following your suggestion, we will revise the paper to clarify the sparse-credit/stop-gradient mechanism, add the new bounded-retrieval and full-scale MSC-QA results, and moderate efficiency and novelty wording.
> > >
> > > ### On the overall recommendation
> > > We sincerely appreciate your constructive and detailed feedback, which has meaningfully improved our paper. The methodological concerns raised in the original review have been addressed, and we have added new evidence on the two remaining points:  a bounded-efficiency variant and a full-scale long-range fidelity evaluation. Given that all raised concerns have been substantially addressed, we would be grateful if the reviewer would consider updating the overall recommendation.

---

### Official Review · Reviewer_Svrw · 2026-03-13

**Soundness:** 3
**Presentation:** 3
**Significance:** 3
**Originality:** 4
**Overall Recommendation:** 5
**Confidence:** 3

**Summary:**

This paper proposes a novel technique for handling extremely long, multiple topically diverse contexts in multi-turn conversations that may span multiple sessions. Naive methods involve truncation beyond the last k turns or summarising the long context. A recently proposed technique, named In-context Autoencoder (ICAE) (Ge et al 2024), trains an autoencoder to compress a long context into a memory slot. When applied to a multi-turn scenario, ICAE would re-encode the entire conversation history, which may be inefficient and may not capture multiple topical threads in a single memory slot.
This paper proposes Context Driven Incremental Compression (CDIC), an extension of ICAE. First, instead of maintaining a single memory slot, they propose maintaining a dynamic number of slots, $\mathcal{M}_t$, that are determined by evolving context. At each turn, all slots similar to the incoming query are selected as compressed context and response generation is conditioned on it. Next, at each turn t, a new memory slot $Z_t$ is computed, conditioned on the selected compressed context $\mathcal{R}_t$, the input query $q_t$, and the generated response $r_t$. Dynamic memory is updated by replacing the existing most similar memory slot with  $Z_t$. If the query initiates a new topical thread (i.e., no existing memory slot has similarity with the query above the threshold), the new memory slot is simply added without removing anything from the dynamic set.  The authors propose to finetune the ICAE autoencoder via truncated backpropagation through the turns.

**Compliance With Llm Reviewing Policy:**

Affirmed.

**Final Justification:**

I had two main concerns -- missing baselines and missing metrics. Both of these have been addressed in the rebuttal.

**Key Questions For Authors:**

1. ICAE uses reconstruction loss for training the autoencoder. Would adding such a loss term help?
2. Which model is used for creating summaries? If it is the same as generator, how do you handle contexts larger than the max tokens that the model supports?

3. Line 146, left column - “Under typical module such as vanilla self–attention, the cumulative cost over an T -turn dialogue is  $O(T^3L^2)$ - the authors cite a survey paper, but I couldn’t find this results. Besides, for a sequence of $TL$ tokens, attention is quadratic $O(T^2L^2)$. I am not sure why the complexity is cubic in the number of turns $T$.

**Limitations:**

Yes

**Strengths And Weaknesses:**

Strengths

1. The paper makes a critical observation that in extremely long multi-turn conversations, there could be multiple topical threads, and a single memory slot in ICAE fails to capture that.

2. The proposed idea of keeping a set of memory slots is technically sound, and the authors try to show empirically that it works better than the other baselines in practice, though I have some reservations about the experimental section.

3. The paper formulates the problem clearly and is well written in general. However, the authors assume readers know about ICAE, which may not be the case, and readers may find it difficult to understand CDIC without knowing ICAE. Since the paper uses ICAE as the starting point, the authors should describe its architecture and training details in the Preliminaries section. For example,


Weaknesses

While I like the overall solution to handle the long context and I find the problem relevant as well,  I have concerns with the experimental section.
My major concerns are the missing baselines and the appropriateness of the metrics used to evaluate the baselines and CDIC.

Metrics: BLEU and RougeL measure lexical overlap with the ground truth. However, they may not correlate with correctness. Further, the best method achieves a BLEU score of 2.3%, which is extremely low, and it is unclear whether this difference is statistically significant.
On the other hand, LLM as Judge - prompting a powerful LLM to evaluate the generated response may semantically compare the generated response with the ground truth. Without an appropriate metric, it is unclear whether the proposed method is empirically superior to the baselines.


Missing baselines: Recently, many approaches have been proposed to compress the context. I am listing a few of them below. I would urge the authors to compare against them as well.

1. InfLLM: Training-Free Long-Context Extrapolation for LLMs with an Efficient Context Memory
2. LLMLingua: Compressing Prompts for Accelerated Inference of Large Language Models

3. AutoCompressors: Adapting Language Models to Compress Contexts

4. Dynamic RAG  -- authors chose a fixed number of chunks (5) when retrieving from the historical context, whereas CDIC applies a threshold to decide on the number of memory slots. It looks like an unfair comparison. I would request that authors show results for 10 and 20 chunks, as well as a variation in which the chunks are dynamically selected based on a threshold. You may treat threshold as a hyper-parameter and select the best based on your train / dev data.

---

> ### Author Rebuttal · Authors · 2026-03-31
>
> ### 1) Comparison should include stronger retrieval/compression baselines
>
> > Thank you for this suggestion. **AutoCompressor is already in Table 1**.  We additionally evaluate **RAG10, RAG20, threshold-based dynamic RAG, LLMLingua and InfLLM** under the same setting; the retrieval threshold is chosen on the dev set. We highlight that **ours remains best on both datasets**.
> >
> > **MSC**
> >
> > | Model | PPL ↓ | BLEU ↑ | R-L ↑ | R-1 ↑ | R-2 ↑ |
> > | --- | --- | --- | --- | --- | --- |
> > | RAG5 | 35.530 | 0.008 | 0.110 | 0.148 | 0.014 |
> > | RAG10 | 34.630 | 0.011 | 0.110 | 0.157 | 0.014 |
> > | RAG20 | 31.179 | 0.013 | 0.111 | 0.156 | 0.014 |
> > | RAG_threshold | 40.057 | 0.011 | 0.109 | 0.157 | 0.013 |
> > | LLMLingua | 36.211 | 0.012 | 0.105 | 0.157 | 0.011 |
> > | InfLLM | 27.329 | 0.016 | 0.118 | 0.161 | 0.019 |
> > | **Ours** | **8.431** | **0.023** | **0.160** | **0.205** | **0.037** |
> >
> > **REALTALK**
> >
> > | Model | PPL ↓ | BLEU ↑ | R-L ↑ | R-1 ↑ | R-2 ↑ |
> > | --- | --- | --- | --- | --- | --- |
> > | RAG5 | 26.789 | 0.020 | 0.103 | 0.151 | 0.015 |
> > | RAG10 | 27.311 | 0.019 | 0.102 | 0.148 | 0.014 |
> > | RAG20 | 27.646 | 0.019 | 0.102 | 0.148 | 0.014 |
> > | RAG_threshold | 28.238 | 0.009 | 0.101 | 0.147 | 0.013 |
> > | LLMLingua | 31.067 | 0.008 | 0.092 | 0.131 | 0.011 |
> > | InfLLM | 29.995 | 0.013 | 0.099 | 0.141 | 0.014 |
> > | **Ours** | **9.789** | **0.035** | **0.134** | **0.176** | **0.030** |
> ### 2)  BLEU/RougeL measure lexical overlap. However, they may not correlate with correctness
>
> > Thank you for your insightful review. To address this directly, we add **MSC-QA**, a targeted reference-tracking QA evaluation on 50 MSC examples. We use GPT-4o to build QA pairs whose answers require earlier contextual mentions, and score them with the same GPT-4o protocol as **LongMemEval (Appendix L)**. Under MSC-QA, our method performs best, consistent with the QA and the lexical-overlap results.
> >
> >**MSC-QA (targeted reference-tracking QA)**
> >
> > | Model | Accuracy ↑ | PPL ↓ | BLEU ↑ | R-L ↑ | R-1 ↑ | R-2 ↑ |
> > | --- | --- | --- | --- | --- | --- | --- |
> > | Full Prompting | 0.04 | 25.195 | 0.017 | 0.124 | 0.162 | 0.026 |
> > | ICAE (incremental) | 0.00 | 877.317 | 0.008 | 0.066 | 0.077 | 0.007 |
> > | ICAE (oneshot) | 0.00 | 32.296 | 0.012 | 0.097 | 0.131 | 0.013 |
> > | InfLLM | 0.12 | 9.185 | 0.035 | 0.155 | 0.199 | 0.054 |
> > | **Ours** | **0.18** | **5.452** | **0.060** | **0.213** | **0.263** | **0.079** |
> >
> > We also highlight three complementary analyses already in the paper: (i) **Appendix E** shows many reference responses depend on distant prior context (lexical-overlap metrics become at least partially informative); (ii) closed-loop rollout (**Table 2**) tests long-horizon stability under self-generated history (probing error accumulation across turns); (iii) LongMemEval QA and qualitative fact-recall analyses (**Appendix L, O**) test long-context recall.
> ### 3)  The best method achieves a BLEU score of 2.3%; is the improvement meaningful and statistically significant?
>
> > Low absolute BLEU is expected in **open-ended dialogue**, where many responses are valid. What matters is whether gains are **consistent across evaluations**. Here, BLEU gains are accompanied by improvements in **PPL/ROUGE, closed-loop behavior, and long-context QA**. We also report a **paired t-test on seed-wise differences** in **Appendix G.2**. The gains are **statistically significant** across all reported metrics (**p < 0.05**).
> ### 4) Whether adding reconstruction loss would further help dialogue compression
>
> > Although we do not add reconstruction loss during dialogue adaptation, our compressor is initialized from a pretrained ICAE checkpoint and already inherits reconstruction-based training. We also tested explicit reconstruction loss during dialogue fine-tuning, but it substantially increased memory usage and led to **OOM**. More importantly, our goal is not exact text reconstruction, but preserving information useful for future response generation under repeated retrieval and write-back.
> ### 5) Summarization baseline details and handling of overlong histories are underspecified
>
> > Thank you for your careful reading. We will clarify that the summarization baseline uses the same Llama2-7B-chat backbone for a controlled comparison. For overlong histories, it maintains a running recursive summary: at each step, the model summarizes the previous summary together with newly observed turns, and the updated summary becomes the future textual history.
> ### 6) The $O(T^3 L^2)$ complexity statement should be derived explicitly
>
> > We will clarify that this is the **cumulative** cost over the full dialogue. If each turn contributes $L$ tokens, then at turn $t$ the prompt length is $tL$, so vanilla self-attention costs $O((tL)^2)=O(t^2L^2)$ at that turn. Summing over $t=1…T$ gives $O(L^2 Σ_t t^2)=O(T^3L^2)$.
> ### 7) ICAE should be explained more clearly in the preliminaries
>
> > Thank you. We will revise Section 2.3 to make ICAE clearer.

---

> > ### Author Rebuttal · Reviewer_Svrw · 2026-04-01
> >
> > I had two main concerns -- missing baselines and missing metrics. Both of these have been addressed in the rebuttal.

---

> > > ### Author Response · Authors · 2026-04-01
> > >
> > > Thank you very much for the thoughtful follow-up and for taking the time to revisit our rebuttal. We sincerely appreciate your positive feedback and are glad that our additional experiments and clarifications addressed all your concerns. Your comments were very helpful in improving both the presentation and evaluation of the paper.

---

### Official Review · Reviewer_k3Xd · 2026-03-13

**Soundness:** 3
**Presentation:** 3
**Significance:** 3
**Originality:** 3
**Overall Recommendation:** 4
**Confidence:** 3

**Summary:**

The authors proposed Context-Driven Incremental Compression (C-DIC), a framework for efficient multi-turn dialogue modeling. Instead of treating a dialogue as a single linear sequence and repeatedly encoding the full history, C-DIC models the conversation as a set of interleaved topical threads. It maintains a compact memory of compressed thread states. At each turn, the system retrieves the relevant thread states, generates a response using a frozen LLM backbone, and incrementally updates the memory (either by revising the existing thread or inserting a new one) using a gradient-free write-back rule. To train the trainable compressor effectively, the authors introduce a retrieval-aware truncated backpropagation through time (ra-TBPTT) mechanism that routes gradients only along the actively utilized memory paths. The manuscript assesses the concept of using thread-aware compressed memory as an alternative to repeatedly processing the full dialogue history in long multi-turn conversations.

**Compliance With Llm Reviewing Policy:**

Affirmed.

**Final Justification:**

I will keep my positive score.

**Key Questions For Authors:**

1. **Baseline fairness:** Could you clarify whether AutoCompressor was re-trained or adapted on MSC/REALTALK under a comparable data/computation budget, or whether the reported numbers follow the original released setup? This matters because part of the gain may otherwise come from task-specific adaptation rather than from the proposed memory mechanism alone.
2. **Hyperparameter sensitivity:** The appendix provides an ablation on τ, which is helpful. However, it would still be useful to understand the sensitivity to α and, more broadly, how these hyperparameters affect response quality, memory growth, and the latency-quality trade-off across datasets.
3. **Thread assumption validity:** Can you provide an analysis (qualitative or quantitative) of how the model handles challenging realistic dialogue phenomena, such as *topic fusion* (where an utterance depends on multiple distinct threads) or *gradual topic drift*?
4. **Backbone context limits:** Given that Llama2-7B has a 4K context window, the poor performance of full prompting is expected. How do you anticipate the quality comparison (excluding latency/memory benefits, which are clear) would change if evaluated on a modern LLM with a natively massive context window (e.g., 128K+)?

**Limitations:**

The authors have not adequately discussed the limitations of their thread-based memory assumption. The rigid “insert-or-revise” update rule may struggle with complex conversational dynamics such as topic fusion or smooth topic drift. It is suggested to add a discussion of these failure modes and how erroneous memory updates might accumulate over long conversations.

**Strengths And Weaknesses:**

**Strengths:**

The proposed method is technically sound regarding its efficiency claims. By conditioning on fixed-size compressed states and using a gradient-free update rule during inference, the per-turn computational cost is effectively decoupled from the total dialogue length, resulting in highly stable latency even for conversations spanning hundreds of turns.

The paper is generally well-structured, and the motivation for moving from sequence-level compression to thread-level memory is clearly articulated.

The work addresses a highly relevant problem—managing context in long-horizon LLM deployments. The proposed latency/memory trade-offs provide clear practical and systems value for real-world dialogue applications.

Reframing dialogue history compression from a flat, one-shot sequence to an incremental, thread-aware memory bank is a novel and intuitively appealing inductive bias. The retrieval-aware TBPTT training scheme also provides a meaningful algorithmic contribution by aligning credit assignment with the retrieved memory path rather than the full dialogue history.

**Weaknesses:**

There are two main concerns regarding empirical soundness. First, the fairness of the baseline comparison is somewhat unclear. The proposed method fine-tunes an ICAE checkpoint on the target datasets (MSC and REALTALK); it is unclear if strong baselines like AutoCompressor received a comparable adaptation budget. Second, the experiments rely on the Llama2-7B backbone, which has a limited 4K context window. The underperformance of the "full prompting" baseline might largely stem from this backbone limitation rather than a fundamental flaw in full prompting, which performs much better on modern long-context open-source models. Furthermore, the memory routing and update policy still depends on several hand-designed choices, including the retrieval threshold τ, recency decay α, argmax fallback, and the insert-vs-revise write-back rule, while the robustness of these design choices is only partially validated.

Algorithm 1 omits the explicit similarity-scoring step, making the retrieve/write-back logic slightly difficult to follow on the first read. A more explicit pseudocode would improve clarity.

The core assumption that open-domain dialogues can be cleanly decomposed into separable, orthogonal threads may be too rigid. In realistic conversations, this thread-based abstraction may break down in at least two common cases. First, topic fusion: a single utterance may simultaneously depend on multiple earlier threads (e.g., combining a previously discussed travel budget with a separate laptop-purchase decision). In such cases, C-DIC’s insert-or-revise update rule faces a dilemma: it can either revise one slot, leaving the other dependency unmodeled, or create a new slot, which may fragment or duplicate the underlying memory structure. Second, gradual topic drift: conversations often evolve smoothly rather than via clear topic switches (e.g., from book recommendations, to a historical event mentioned in a book, to food habits in that historical period). Without a clear boundary, a fixed threshold τ may either collapse these evolving turns into a single increasingly overloaded slot—raising the risk of forgetting—or split them too aggressively, weakening cross-turn coherence.

---

> ### Author Rebuttal · Authors · 2026-03-31
>
> ### 1) Fairness of the adaptation budget across learned baselines
>
> >Thank you for raising the fairness concern. We clarify that our model, C-DIC, and all learned compression baselines, AutoCompressor and ICAE, are finetuned only on MSC under the same 2-epoch setting. Thus, evaluations on REALTALK and LongMemEval are fully zero-shot and do not involve any additional in-domain tuning. Instead, they measure how well the compression and memory mechanisms transfer across datasets. We will make this protocol explicit in the revision.
>
> ### 2) Whether full prompting is disadvantaged by the 4K context limit of the shared 7B backbone
>
> >Thank you for the helpful suggestion. The purpose of the shared 7B setup is to enable a **controlled comparison of context-management methods** under the same generator. To address this concern, we added evaluation against *Full Prompting* with a backbone of Llama-3.1-8B-Instruct (128k context) on both MSC and REALTALK.
> >
> >**MSC**
> >
> > | Model | PPL ↓ | BLEU ↑ | R-L ↑ | R-1 ↑ | R-2 ↑ |
> > | --- | --- | --- | --- | --- | --- |
> > | Full Prompting | 24.244 | 0.015 | 0.116 | 0.157 | 0.017 |
> > | Ours | **11.211** | **0.025** | **0.157** | **0.199** | **0.036** |
> >
> > **REALTALK**
> >
> > | Models | PPL ↓ | BLEU ↑ | R-L ↑ | R-1 ↑ | R-2 ↑ |
> > | --- | --- | --- | --- | --- | --- |
> > | Full Prompting | 29.444 | 0.017 | 0.109 | 0.158 | 0.018 |
> > | **Ours** | **15.471** | **0.022** | **0.129** | **0.166** | **0.028** |
> >
> > Our method still outperforms Full Prompting even with a stronger long-context backbone. We will add this comparison in the revision.
> ### 3)  Sensitivity of response quality, memory growth, and retrieval cost to the threshold τ and decay α
>
> > Thank you for the valuable suggestion. We now add all the analyses including: an **α-sensitivity analysis** alongside the appendix results for τ, and report not only response quality but also **average total and retrieved slot counts** to reflect **memory growth** and **retrieval usage**. The main pattern is consistent across MSC and LongMemEval: performance is stable over a reasonable range, while overly strict retrieval (τ=.9) sharply increases memory size because the model stops revising existing slots and keeps appending new ones. Moderate recency bias is also robust, and on LongMemEval even improves accuracy while reducing retrieval breadth.
> >
> >**MSC — Threshold (τ) sweep**
> >
> > | Threshold | PPL ↓ | BLEU ↑ | R-L ↑ | R-1 ↑ | R-2 ↑ | Total Slots | Retrieved Slots |
> > | --- | --- | --- | --- | --- | --- | --- | --- |
> > | 0.6 | 8.325 | 0.028 | 0.157 | 0.208 | 0.037 | 3.022 | 1.000 |
> > | 0.7 | 8.383 | 0.027 | 0.156 | 0.207 | 0.036 | 3.022 | 1.000 |
> > | 0.8 | 8.427 | 0.030 | 0.160 | 0.206 | 0.040 | 3.501 | 1.015 |
> > | 0.9 | 12.202 | 0.022 | 0.139 | 0.190 | 0.027 | 30.026 | 1.050 |
> >
> > **MSC — Decay rate (α) sweep**
> >
> > | Decay rate | PPL ↓ | BLEU ↑ | R-L ↑ | R-1 ↑ | R-2 ↑ | Total Slots | Retrieved Slots |
> > | --- | --- | --- | --- | --- | --- | --- | --- |
> > | 0 | 8.399 | 0.025 | 0.160 | 0.207 | 0.036 | 3.150 | 1.005 |
> > | 0.05 | 8.427 | 0.030 | 0.160 | 0.206 | 0.040 | 3.501 | 1.015 |
> > | 0.1 | 8.353 | 0.026 | 0.157 | 0.207 | 0.038 | 3.022 | 1.000 |
> >
> > **LongMemEval — Threshold (τ) sweep**
> >
> > | Threshold | Accuracy↑ | Total Slots | Retrieved Slots |
> > | --- | --- | --- | --- |
> > | 0.6 | 0.104 | 1.004 | 1.002 |
> > | 0.7 | 0.114 | 1.362 | 1.0642 |
> > | 0.8 | 0.116 | 18.762 | 1.980 |
> > | 0.9 | 0.114 | 160.658 | 1.242 |
> >
> > **LongMemEval — Decay rate (α) sweep**
> >
> > | Decay rate | Accuracy↑ | Total Slots | Retrieved Slots |
> > | --- | --- | --- | --- |
> > | 0 | 0.098 | 23.920 | 2.810 |
> > | 0.05 | 0.116 | 18.762 | 1.980 |
> > | 0.1 | 0.120 | 5.014 | 1.478 |
>
> ### 4) Algorithm 1 needs explicit retrieval-scoring and write-back steps
> >Thank you for your detailed suggestions on presentation. In the revision, we will rewrite Algorithm 1 to first compute the similarity scores $s_i = S(q_t, Z_i)$, then derive $R_t$, $δ_t$, and $j_t$, and finally perform generate → compress → write-back. This should make the retrieve/write-back logic much easier to follow on first read.
>
> ### 5) Whether a thread-based memory is too rigid under topic fusion and gradual topic drift
> >Thank you for your critical concern. Since dialogues exhibit varying topic dynamics, C-DIC does not assume explicit, human-defined topic categories that cleanly partition a dialogue into orthogonal topics. Rather, the *threading* in our paper refers to **implicit context groupings** into memory latents. Our method **adaptively** organizes dialogue turns into threads based on their contextual dynamics, thereby handling gradual topic drift. For topic fusion, single-slot revision does not leave other retrieved memory states unmodeled since our compressor encodes all retrieved states into a new state representing all relevant topics. Moreover, creating additional slot for an existing topic is harmless since C-DIC will retrieve and use all contextually relevant slots even with a recency bias.

---

### Decision · Program_Chairs · 2026-04-30

**Decision:**

Accept (regular)

**Comment:**

The paper argues that long multi-turn chat is inefficient if you either keep feeding the whole history every turn, truncate it, or rely on static summaries/compressors. Their claim is that existing latent compressors are mostly built for static settings and break under repeated multi-turn rollout.

Their proposed method, Context-Driven Incremental Compression (C-DIC), treats a dialogue as a set of evolving latent “threads.” At each turn, it retrieves relevant memory slots, generate/compress conditioned on those retrieved slots, write back by either revising the best-matching slot or inserting a new one. Training uses a retrieval-aware truncated BPTT that routes gradient mainly along the updated thread path instead of the whole conversation history.

Experiments show C-DIC beats full prompting, truncation, summarization, RAG, AutoCompressor, and ICAE variants on MSC and REALTALK, with good latency control.

While all reviewers provided positive feedbacks, the main concerns raised in the initial reviews were threefold. First, reviewers questioned whether the empirical evaluation was sufficiently broad and fair, particularly with respect to stronger retrieval/compression baselines and the comparison against full prompting under a limited-context backbone. Second, several reviewers noted that the main dialogue evidence relied heavily on PPL/BLEU/ROUGE, and requested more direct evaluation. Third, there were concerns about the precision of some technical claims, like the clarity of the training/inference distinction. The rebuttal addressed these concerns well.

Overall, I find this to be a solid contribution on an important problem.  The final version should moderate some claims, tighten the presentation, and incorporate the new rebuttal evidence directly into the paper.